# Two-dimensional charge order stabilized in clean polytype heterostructures

Suk Hyun Sung [1], Noah Schnitzer[2,3], Steve Novakov [4], Ismail El Baggari[5,6], Xiangpeng Luo[4], Jiseok Gim [1], Nguyen M. Vu[1], Zidong Li [7], Todd H. Brintlinger [8], Yu Liu [9], Wenjian Lu [9], Yuping Sun [9,10,11], Parag B. Deotare [7,12], Kai Sun [4], Liuyan Zhao [4], Lena F. Kourkoutis [3,13], John T. Heron [1,12] & Robert Hovden [1,12]✉

Compelling evidence suggests distinct correlated electron behavior may exist only in clean 2D materials such as $1T\text{-}TaS_2$. Unfortunately, experiment and theory suggest that extrinsic disorder in free standing 2D layers disrupts correlation-driven quantum behavior. Here we demonstrate a route to realizing fragile 2D quantum states through endotaxial polytype engineering of van der Waals materials. The true isolation of 2D charge density waves (CDWs) between metallic layers stabilizes commensurate long-range order and lifts the coupling between neighboring CDW layers to restore mirror symmetries via interlayer CDW twinning. The twinned-commensurate charge density wave (tC-CDW) reported herein has a single metal–insulator phase transition at ~350 K as measured structurally and electronically. Fast in-situ transmission electron microscopy and scanned nanobeam diffraction map the formation of tC-CDWs. This work introduces endotaxial polytype engineering of van der Waals materials to access latent 2D ground states distinct from conventional 2D fabrication.

[1] Department of Materials Science and Engineering, University of Michigan, Ann Arbor, MI 48109, USA. [2] Department of Materials Science and Engineering, Cornell University, Ithaca, NY 14853, USA. [3] Kavli Institute at Cornell for Nanoscale Science, Ithaca, NY 14853, USA. [4] Department of Physics, University of Michigan, Ann Arbor, MI 48109, USA. [5] Department of Physics, Cornell University, Ithaca, NY 14853, USA. [6] Rowland Institute at Harvard, Cambridge, MA 02142, USA. [7] Electrical and Computer Engineering Department, University of Michigan, Ann Arbor, MI 48109, USA. [8] Materials Science and Technology Division, U.S. Naval Research Laboratory, Washington, D.C. 20375, USA. [9] Key Laboratory of Materials Physics, Institute of Solid State Physics, Chinese Academy of Sciences, 230031 Hefei, P. R. China. [10] High Magnetic Field Laboratory, Chinese Academy of Sciences, 230031 Hefei, P. R. China. [11] Collaborative Innovation Centre of Advanced Microstructures, Nanjing University, 210093 Nanjing, P. R. China. [12] Applied Physics Program, University of Michigan, Ann Arbor, MI 48109, USA. [13] School of Applied and Engineering Physics, Cornell University, Ithaca, NY 14853, USA. ✉email: hovden@umich.edu

Charge density waves (CDW) are an emergent periodic modulation of the electron density that permeates a crystal with strong electron-lattice coupling[1–4]. TaS₂ and TaS$_{x-}$Se$_{2-x}$ host several CDWs that spontaneously break crystal symmetries, mediate metal–insulator transitions, and compete with superconductivity[1,5–7]. These quantum states are promising candidates for novel devices[8–11], efficient ultrafast non-volatile switching[12,13], and suggest elusive chiral superconductivity[14,15]. Law and Lee recently called for pristine 2D CDW syntheses to access exotic spin-liquid states in 1T-TaS₂[16]. Unfortunately, extrinsic and thermal disorder in free standing 2D layers degrades correlation-driven quantum behavior[17,18] and clean 2D charge density waves or superconductivity are near absent[19]. Room temperature access to spatially-coherent charge density waves (e.g., commensurate states) and clean 2D confinement could enable a paradigm shift toward device logic and quantum computing.

Here, we show the critical temperature for spatially-coherent, commensurate charge density waves (C-CDW) in 1T-TaS₂ can be raised to well above room temperature (~150 K above the expected transition) by synthesizing clean (minimal impurities or defects) interleaved 2D polytypic heterostructures. This stabilizes a collective insulating ground state (i.e., C-CDW) not expected to exist at room temperature. We show the formation of these spatially-coherent states occurs when 2D CDWs are confined between metallic prismatic polytypes. Metallic layers screen impurity potentials to suppress the nearly-commensurate (NC-CDW) phase. At the same time, interleaving disables interlayer coupling between CDWs to ensure an unpaired electron in each 2D supercell. This raises the critical temperature of the C-CDW and forms out-of-plane twinned commensurate (tC) CDWs as revealed by scanned nanobeam electron diffraction. These results demonstrate polytype engineering as a route to isolating 2D collective quantum states in a well-defined extrinsic environment with identical chemistry but distinct band structure.

Layered TaS₂ polytypes (Supp. Fig. S2) are archetypal hosts to anomalous electronic properties associated with the formation of CDWs. The Ta coordination to six chalcogens dramatically changes its behavior. Prismatic coordination (Pr) found in the stable 2H polytype is metallic, even below the CDW onset around 90 K, and becomes superconducting around 0.5 K (enhanced to 2.2 K when thinned[5]). Octahedral (Oc) coordination found in the metastable 1T polytype has inversion symmetry and exhibits three distinct, salient CDW phases: commensurate (C), nearly-commensurate (NC), and incommensurate (IC). An intermediate triclinic phase has also been reported[20–22]. At room temperature, the conductive NC-CDW is generally accepted as a C-CDW with short range order[23–27] that permits electron transport along regions of discommensuration[28–30]. Below ~200 K, the CDW wave vector locks into ~13.9° away from the reciprocal lattice vector (Γ–M) to become a C-CDW that achieves long-range order with a $\sqrt{13} \times \sqrt{13}$ supercell[1,31]. This reduction of crystal symmetry gaps the Fermi-surface and the commensurate phase becomes Mott insulating[1,32,33]. Above 352 K, the CDW wave vector aligns along the reciprocal lattice vector and becomes the disordered IC-CDW phase.

## Results

**Twinned commensurate charge density waves.** The tC-CDW phase reported herein has distinct out-of-plane charge order—illustrated in Fig. 1a (See also: Supp. Fig. S1). 2D CDWs reside within Oc-layers sparsely interleaved between metallic Pr-layers. Each CDW is commensurate in one of two degenerate twin states, α-C (blue) or β-C (red) (Fig. 1b, c). The translational symmetry is described by in-plane CDW lattice vectors ($\lambda_{\alpha1}$, $\lambda_{\alpha2}$) and ($\lambda_{\beta1}$, $\lambda_{\beta2}$).

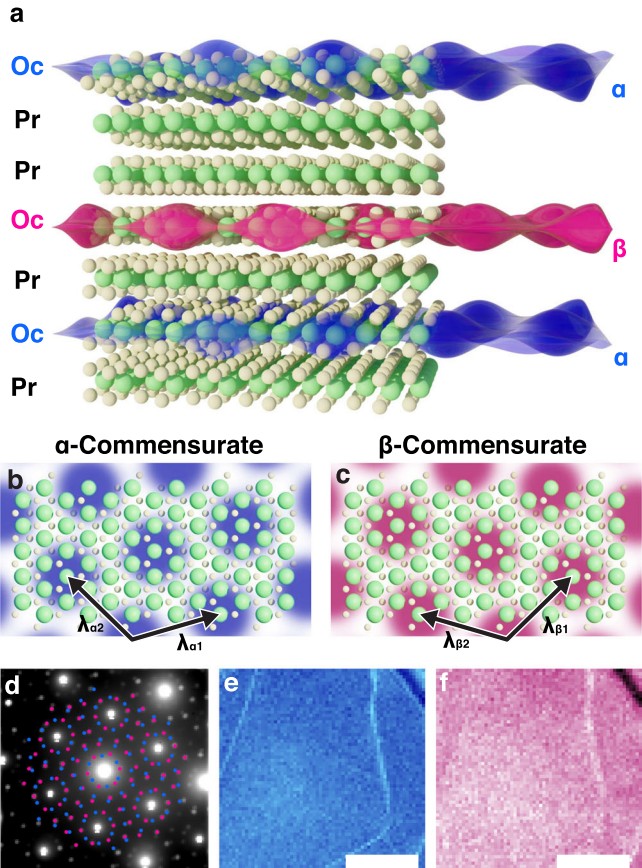

**Fig. 1 Twinned, commensurate CDW at room temperature in ultrathin TaS₂. a** Schematic illustration of room-temperature, out-of-plane twinned, commensurate CDW in 1T-TaS₂. Blue and red overlays depict CDW twins within octahedrally coordinated TaS₂. Metallic prismatic polytypes isolate octahedral layers to stabilize tC-CDWs. **b, c** Twin superlattice structure illustrated for α and β C-CDW, respectively. **d** Average diffraction pattern of twinned, C-CDW state over (870 nm)² field-of-view reveals two sets of superlattice peaks (marked with blue and red; unmarked image available in Supplementary Fig. S12a). **e, f** Nanobeam diffraction imaging from each set of superlattice peaks maps the coexistence of both CDW twins—expected for twinning out-of-plane. Scale bar is 300 nm.

CDWs are a prototypical manifestation of electron-lattice coupling, in which both the electron density and lattice positions undergo periodic modulations to reduce crystal symmetry and lower the electronic energy[34]. The associated periodic lattice distortions (PLD) diffract incident swift electrons into low-intensity superlattice peaks between Bragg peaks[35,36]. Polytypes and stacking order thereof manifests as changes in Bragg peak intensities whereas CDWs produce distinct superlattice peaks[25,35,37] (Supplementary information SI3). Figure 1d shows the position averaged convergent beam electron diffraction pattern (0.55 mrad semi-convergence angle, 80 keV) of the tC-CDW phase at room temperature with α, β superlattice peaks annotated (blue, red). Regularly spaced superlattice peaks and bright first order superlattice peaks are characteristic of C-CDWs and match the tC-CDW peaks.

Both α and β CDW states were mapped over microns of area to reveal a uniform co-existence of both twins when viewed in projection out-of-plane (Fig. 1e, f). This is evidently different from recently reported in-plane twinned CDWs created by femtosecond light pulses[38]. Mapping CDW structure required a pixel array detector with 1,000,000:1 dynamic range[39] allowing Bragg beams to be imaged while still maintaining single electron

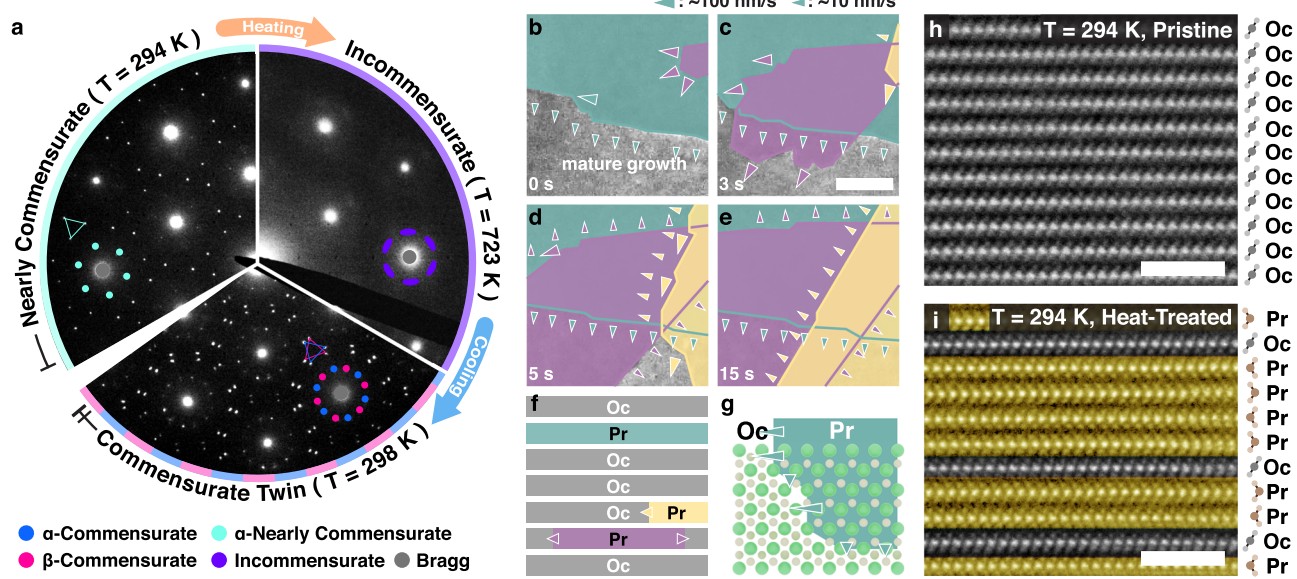

**Fig. 2 Polytype isolation forms 2D CDW layers. a** Pristine 1T-TaS$_2$ at room temperature hosts NC-CDW (left). Upon heating the NC phase gives way to IC-CDW (right) at ~350 K; the transition is normally reversible. Remarkably, heating above ~620 K then cooling stabilizes tC-CDW (bottom). SAED patterns were formed from a 500 nm aperture **b–e** In-situ TEM reveals layer-by-layer octahedral to prismatic polytypic transformations during heat treatment. Multiple polytypic domains (denoted green, purple, and yellow) nucleate and grow simultaneously without interaction (See Supplementary Movie 1). Scale bar is 350 nm. **f** Schematic cross-section of TaS$_2$ during layer-by-layer polytypic transition. **g** Fast and slow transitions occur along $\langle 10\bar{1}0 \rangle$ and $\langle 11\bar{2}0 \rangle$ directions respectively. **h, i** Atomic resolution cross-sectional HAADF-STEM of **h** pristine and **i** heat-treated TaS$_x$Se$_{2-x}$ confirms polytypic transformation. After treatment, prismatic (Pr) layers encapsulate monolayers of octahedral (Oc) layers. Scale bar is 2 nm. A selenium doped sample was imaged to enhance chalcogen visibility.

sensitivity near CDW peaks. A convergent beam electron diffraction (CBED; 0.55 mrad convergence semi-angle, 80 keV) pattern was collected at every beam position across large fields of view—an emerging technique often called 4D-STEM (see "Methods" section)[39–41]. Using this method, the local CDW structure was measured at ~4.6 nm resolution and across >1 µm fields of view. It demonstrates 4D-STEM as an invaluable tool for mapping charge order in materials. Previous approaches to mapping CDWs entailed sparse measurement from a handful of diffraction patterns[38], small-area tracking of atomic displacements[42], or traditional dark-field TEM techniques that result in low resolution and debilitating signal-to-noise ratios[1,43].

**Clean polytype heterostructures**. Thermal treatment reproducibly forms the tC-CDW phase—a process summarized by the in-situ selected area electron diffraction (SAED) in Fig. 2a. Initially, an exfoliated flake of 1T-TaS$_2$ hosts NC-CDWs (Fig. 2a, left) at room temperature with diffuse first-order superlattice peaks (cyan circles) and sharp second order superlattice peaks (cyan triangle). 1T-TaS$_2$ is heated above the reversible phase transition (T$_{NC-IC} \approx 352$ K) into the disordered IC-CDW state, which has characteristic azimuthally diffuse superlattice spots (Fig. 2a, right). Heating continues up to temperatures (~720 K) above the polytype transition (T$_{Oc-Pr} \approx 600$ K[44]), where it remains for several minutes (see "Methods" section). Upon cooling, the system does not return to the expected NC-CDW but instead enters a tC-CDW state with sharp, commensurate first and second order superlattice peaks that are duplicated with mirror symmetry (α, β) (Fig. 2a, bottom). The tC-CDW phase is stable and observable after months of dry storage (RH ~10%) at room temperature. Synthesis was replicated ex-situ in both high-vacuum (<10$^{-7}$ Torr) and inert argon purged gloveboxes, but amorphized in ambient air. The tC-CDW was equivalently synthesized for both TaS$_2$ and TaS$_x$Se$_{2-x}$.

Heating above the polytype transition temperature (T$_{Oc-Pr}$) provokes layer-by-layer transitions from Oc to Pr polytypes instead of a rapid bulk transformation. Figure 2b–e shows in-situ TEM using high-frame-rate (25 fps) microscopy taken at ~710 K. Each colored overlay highlights the growth and formation of a new prismatic polytype domain (raw data in Supp. Fig. S6). Arrows indicate movement of Oc/Pr coordination boundary with a fast-transition up to ~100 nm/s along $\langle 10\bar{1}0 \rangle$ crystal directions and a slow-transition at ~10 nm/s along $\langle 11\bar{2}0 \rangle$ (Fig. 2g). Video of the transformation is striking (Supplementary Movie 1). Domains nucleate and boundaries progress independently between layers as illustrated in Fig. 2b–f. Cooling the sample mid-transition produces a sparsely interleaved polytypic heterostructure.

Atomic resolution cross-section images of pristine and heat-treated samples (Fig. 2h, i, respectively) measured by high-angle annular dark-field (HAADF)-STEM reveal the interleaved polytypic heterostructure. Interleaving isolates monolayers of octahedral (Oc) coordination that host 2D-CDWs in a clean, defect-free environment of metallic prismatic (Pr) layers. Although this system is best described as a sparse interleaving of Oc-layers within many Pr-layers (Fig. 2i), the uncorrelated polytype stacking may permit by chance a small (or even negligible) amount of layers which locally match a 4Hb (or another bulk polytype) unit cell.

The metallic "Pr" layers are hypothesized to screen out-of-plane interactions and impurity potentials to stabilize low-temperature commensurate CDWs at room temperature. As a result, the NC-CDW state no longer exists and a long-range ordered tC-CDW emerges as a stable phase up to ~350 K. This result is radically different from previous reports where free standing ultra-thin 1T-TaS$_2$ degrades long-range order[45] and broadens the NC-CDW phase by lowering T$_{CCDW}$[13,18,46]. However, our observation of 2D commensurate CDWs agrees with a theoretical prediction that commensurate CDWs are more stable in clean monolayer[47].

Understanding the role of disorder requires decoupling intrinsic quantum behavior from extrinsic influences at the surfaces, especially in low dimensions where long-range order becomes more fragile and vulnerable to impurities[17,48,49]. When the disorder strength reaches a certain threshold, the long-range C-CDW phase gives its way to a disordered phase[17]. Here, each 2D 1T-TaS$_2$ CDW is in its native chemical, endotaxial, and unstrained environment. Impurity potentials that pin CDWs[50] and break spatial coherence are mitigated by adjacent metallic Pr-layers. For C-CDWs (in 2D and above) in the presence of sufficiently weak disorder, the charge order remains stable[49]. Additionally, isolating monolayers of 1T-TaS$_2$ ensures an odd number of electrons per unit cell and elongates the Fermi surface out-of-plane—both expected to reduce the electronic energy.

**Isolation of 2D charge density waves.** In the tC-CDW, metallic Pr-layers decouple interlayer CDW interactions to create isolated 2D CDWs with twin degeneracy. Using a phenomenological model we illustrate a kinetic pathway for accessing the tC-CDW. Here, local orientation of the CDW wave vector, $\theta$, is an apt order parameter for describing the breaking of the mirror symmetry in the C $\leftrightharpoons$ IC transition. This provides a simple, minimal phenomenological model to qualitatively capture the formation of twinned CDWs but does not model all remaining components of the complete CDW order parameter (see "Methods" section). A free energy expansion of this order parameter combined with an XY interaction of the CDW wave vector qualitatively reproduces diffraction patterns for IC-CDW and α/β C-CDW. In diffraction, the superlattice peak location and shape encodes the distribution of the CDW order parameter. Simulated diffraction patterns at high temperatures feature first-order superlattice peaks azimuthally broadened by CDW disorder and centered along the reciprocal lattice (Γ–M) direction (Fig. 3b). At low temperature, the superlattice peaks are sharpened by long-range CDW order and located at +13.9° or −13.9° away from Γ–M (Fig. 3a, c). Figure 3d shows distribution of $\theta$ at high (gray) and low (blue, red) temperatures.

When cooled, the system chooses α or β with equal probability. For pristine 1T-TaS$_2$, CDWs couple between layers, twin degeneracy is broken, and no twinning occurs[51]. However, in the absence of interlayer interaction or extrinsic perturbation, each 2D CDW layer quenches randomly into either α or β C-CDW (Fig. 3g) from the high-temperature IC-CDW phase (Fig. 3f)—forming the out-of-plane twinned tC-CDW phase. In projection, the model produces diffraction pattern that resembles experimental SAED of IC-CDW and tC-CDW phases. (Fig. 3f, g insets) In systems with very few CDW layers, one stronger CDW direction can sometimes emerge (Supp. Fig. S12b, c). From our model, out-of-plane twinning occurs for modest cooling rates but we note fast quenching predicts in-plane twinning (Supp. Fig. S10) similar to reported ultrafast optical excitations[38].

**Optical and electrical characteristics of tC-CDW.** Rotational-anisotropy second-harmonic generation (RA-SHG) revealed restoration of twin degeneracy in heat-treated samples. The RA-SHG of pristine sample (Fig. 4b) exhibits a hallmark of the RA-SHG pattern rotated away from the lattice vectors; breaking mirror symmetry due to formation of a single-domain NC-CDW. In contrast, the heat-treated sample's RA-SHG pattern (Fig. 4c) is mirror-symmetric to the crystalline directions and much stronger. Together, this is a strong evidence of equally weighted degenerate α and β states (i.e., tC-CDW) and the emergence of Pr-layers that are mirror symmetric and non-centrosymmetric.

Electronic measurement of the polytypic heterostructure with interleaved CDWs shows a direct tC $\leftrightharpoons$ IC transition at 350 K and

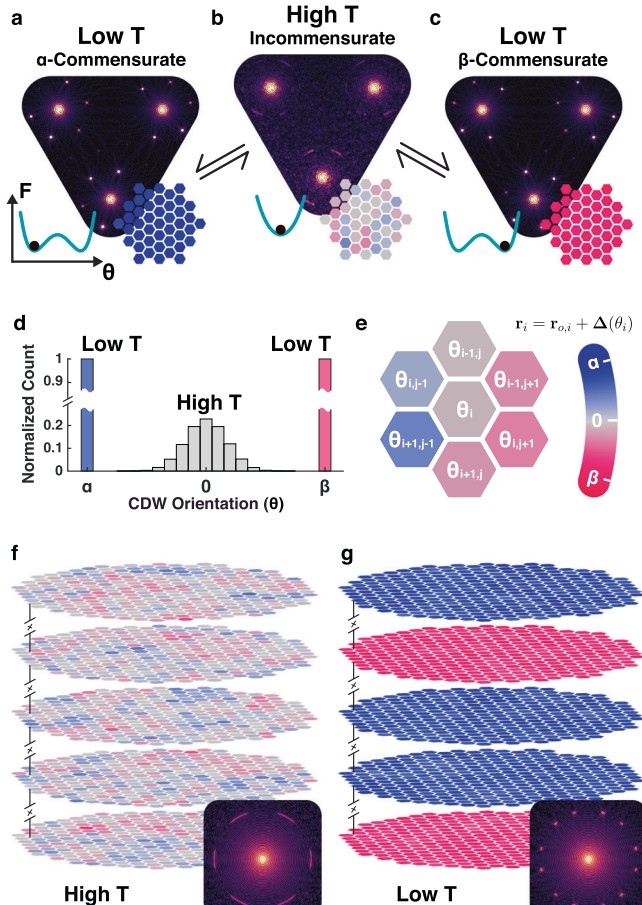

**Fig. 3 Phenomenological model illustrates formation of commensurate CDWs with out-of-plane twin degeneracy. a–c** The CDW wave-vector direction $\theta$ defines an order parameter with degenerate commensurate twins when cooled from IC-CDW phase. Simulated far-field diffraction patterns for **a** α-C, **b** IC, and **c** β-C. The free energy (F) landscape (Inset-left) governs the mean $\theta$ and the real-space distribution (Inset-right). **d** Histogram of $\theta$ shows zero-centered, wide distribution at high temperature. At low temperature, the distribution is narrow and centered at ±13.9° for either twin. **e** Map of local orientation of wave vector ($\theta$) at IC phase. Each hexagonal cell represents $\theta$ at each Ta site. **f** At high temperature $\theta$ is mean centered and disordered, however, **g** at low-temperature each 2D layer converges into either α or β randomly when layers are decoupled. Simulated far-field diffraction patterns of multi-layer system in high temperature (Inset—**f**) and low temperature (Inset—**g**) resembles experimentally observed SAEDs.

disappearance of the disordered NC-CDW phase. Figure 4a shows in-plane resistance vs. temperature measurements of pristine 1T (orange) and heat-treated polytypic heterostructure (blue) are dramatically different. The heterostructure features only one metal–insulator transition at 350 K, whereas pristine 1T-TaS$_2$ exhibits two transitions at ~200 K (C $\leftrightharpoons$ NC) and 350 K (NC $\leftrightharpoons$ IC). Jumps in resistance are a signature of emergent CDW order. At low temperature the metallic Pr-layers dominate in-plane conduction since their resistance is expected to monotonically decrease as measured in bulk material[52]. This hinders the quantification of resistance in individual Oc-layers, however the critical temperatures remain clearly visible.

Repeated in-situ heating–cooling cycles reveal the tC $\leftrightharpoons$ IC transition is reversible and the NC-CDW phase is removed. Note that the intermediate triclinic phase was also not observed. Electronic measurements (Fig. 4a) match structure measurements

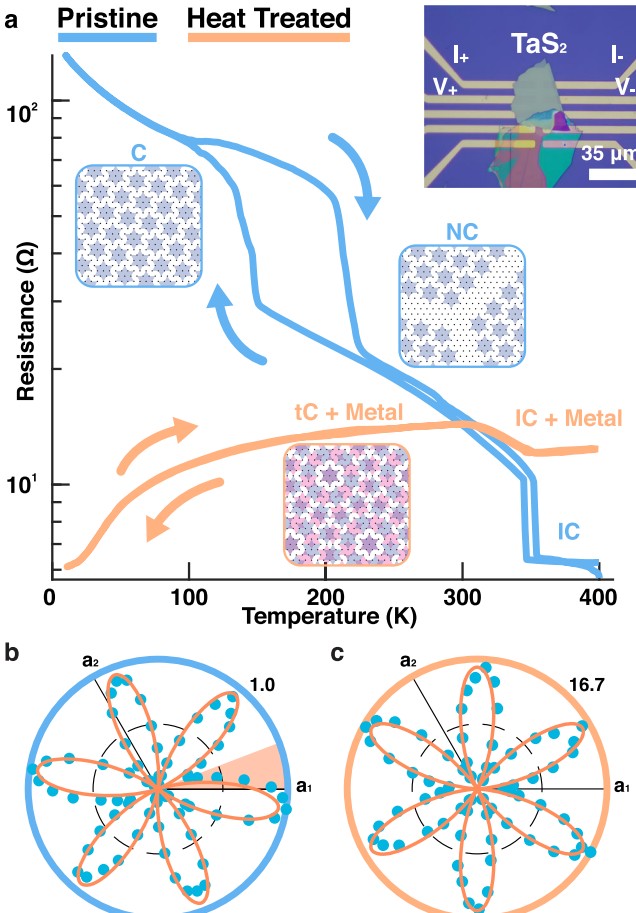

**Fig. 4 Electronic transport of tC-CDW phase transition and reversibility. a** 4-point in-plane resistance measurement as function of temperature for pristine bulk (orange) and heat-treated (blue) TaS₂. Pristine samples show two jumps in resistance for C ⇌ NC and NC ⇌ IC, whereas the heat treated polytypic heterostructures only feature a single, reversible tC ⇌ IC transition at ~350 K corresponding to the enhanced critical temperature for CDW commensuration and disappearance of the NC-CDW. Metallic Pr-layers dominate the overall trend of the resistance measurement, however, the single jump above room-temperature is a distinct feature of the tC-CDW. Schematics represent a simplified CDW structures of each phase. Inset) Optical image of the nanofabricated device. **b** The RA-SHG pattern for pristine 1T samples display a mismatch between the nominal mirror direction and the crystalline direction, indicating the CDW breaks mirror symmetry. **c** After heat treatment, the RA-SHG pattern is symmetric with respect to the crystal, implying equal weights between the α and β states. The SHG intensity also increases with mirror symmetric Pr-layers present.

from in-situ SAED (Fig. 2a) and confirm the insulating commensurate charge order of the tC-CDW phase. The interleaved polytype herein stabilizes coherent electronic states well above (~150 K) the normal critical temperature, and disordered NC-CDW phase is structurally and electronically removed.

## Discussion

In summary, we have established a pathway toward stabilizing latent CDWs with long-range order at room temperature by reducing their dimensions to 2D using clean interleaved polytypes of TaS₂. The metallic layers isolate each 2D CDW to restore twin degeneracy giving rise to an out-of-plane twinned commensurate CDW phase. 4D-STEM proved invaluable for mapping CDW

domains across large fields-of-view (~1 μm) to confirm twin structure while atomic resolution HAADF-STEM revealed the 2D CDW layers within a metallic phase. Both structural and electronic investigations show the disordered NC-CDW phase disappears when a 2D CDW is in a chemically and endotaxially native clean environment. The stabilization of ordered electronic phases with 2D polytype engineering has significant implications for routes to access fragile, exotic correlated electron states.

## Methods

**Electron microscopy.** In-situ 4D-STEM was performed on FEI Titan Themis 300 (80 keV, 0.55 mrad convergence semi-angle) with electron microscope pixel array detector (EMPAD) and DENS Wildfire heating holder. For 4D-STEM, a convergent beam electron diffraction (CBED) pattern was recorded at each beam position using the EMPAD detector. EMPAD's high dynamic range (1,000,000:1) and single electron sensitivity[39] allows simultaneous recording of intense Bragg beams alongside weak superlattice reflections. Virtual satellite dark field images were formed by integrating intensities from all satellite peaks at each scan position.

In-situ TEM revealed layer-by-layer polytypic transition. The polytype difference is indiscernible in low-magnification with HAADF-STEM, as two polytypes have equal density. However, the change in bond coordination at polytype boundaries provide visible coherent contrast in TEM. The in-situ movie and SAED patterns were taken with JEOL 2010F (operated at 200 keV) with Gatan Heater Holder and Gatan OneView Camera.

Cross-sectional HAADF-STEM images were taken on JEOL 3100R05 (300 keV, 22 mrad) with samples prepared on FEI Nova Nanolab DualBeam FIB/SEM.

**Preparation of TEM samples.** Both samples for 4D-STEM and TEM were prepared by exfoliating bulk 1T-TaS₂ and 1T-TaSₓSe₂₋ₓ crystals onto poly-dimethylsiloxane (PDMS) gel stamp. The sample was then transferred to TEM grids using home-built transfer stage. Silicon nitride membrane window TEM grid with 2 μm holes from Norcada and DENS solutions Wildfire Nano-chip grid were used. From optical contrast and CBED patterns, the samples (Figs. 1 and 2) were estimated to be 20–50 nm thick[53,54].

**Synthesis of 1T-TaS₂ crystal.** High-quality single crystals of 1T-TaS₂ and 1T-TaSₓSe₂₋ₓ (x ≈ 1) were grown by the chemical vapor transport method with iodine as a transport agent. Stoichiometric amounts of the raw materials, high-purity elements Ta, S, and Se, were mixed and heated at 1170 K for 4 days in an evacuated quartz tube. Then the obtained TaSₓSe₂₋ₓ powders and iodine (density: 5 mg/cm³) were sealed in another longer quartz tube, and heated for 10 days in a two-zone furnace, where the temperature of source zone and growth zone was fixed at 1220 K and 1120 K, respectively. A shiny mirror-like sample surface was obtained, confirming their high quality. All CDW characterization was done on 1T-TaS₂; Se-doped sample was used only for polytype characterization in cross-sectional HAADF-STEM (Fig. 2h, i and Fig. S7) and cross-sectional SAED (Supp. Fig. S8).

**Thermal synthesis of tC-CDW in TaS₂.** Interleaved 2D TaS₂ polytypes were synthesized by heating 1T-TaS₂ to 720 K in high vacuum (<10⁻⁷ Torr) or in an argon purged glovebox[55]. 1T-TaS₂ was held at 720 K for ~10 min, then brought down to room temperature. Once the interleaved polytype is fully established, the tC-CDW becomes stable electronic state up to 350 K. Synthesis is sensitive the duration held at high temperature. Notably, if the sample is not held long enough at elevated temperature, a mixed NC-C twin phase was reached (See Supplementary Fig. S8), due to sparse distribution of Pr-layers encapsulating multilayer 1T-TaS₂. Both slow (~1 K/s) and fast (~10 K/s) cooling rate produced out-of-plane twinning.

**Device fabrication and electronic measurement.** Sample lithography was done with a standard SPR-220 based process using a Heidelberg μPG 501 maskless exposure tool to directly expose samples. The bottom contact pattern was exposed on a silicon wafer with 500 nm SiO₂. To limit damage and bending to the TaS₂ flakes, the bottom contact liftoff pattern was first ion beam etched with Ar to a 60 nm depth. The trenches were then back-filled with 10 nm Ti and 50 nm Au, deposited in an e-beam evaporator at 3 Å/s, and excess metal was lifted-off using acetone/isopropanol sonication of the wafer. The resulting contacts are level to the wafer surface to within 5 nm, as confirmed by AFM. Before flake exfoliation, the bottom contacts were gently polished using 100 nm micropolishing film to remove liftoff fingers, which can extend up to 30 nm above the wafer surface. Each bottom contact pattern includes six 5 μm striplines spaced 5 μm apart, and eight radial pads of approximately 500 × 500 μm for top contact fan-out.

Resistance vs. temperature measurements were performed in a Quantum Design Dynacool PPMS using a standard sample puck and an external Keithley 2400 series source meter. The sample was adhered to the puck backplane with silver paint, and contacts were wire bonded to the puck channel pads using 50 μm Au wire. To ensure sample thermalization, a baffle rod with an Au-coated sealing disk hovering <1 cm above the sample was inserted into the PPMS bore, and the

heating and cooling rate was restricted to <2 K/min. Depending on sample characteristics, between 20–200 μA current was sourced for four wire measurements. The current/voltage limits were chosen to keep electric fields below 10 kV/cm to avoid sample breakdown, as well as to keep current densities below $10^5$ A/cm$^2$ and prevent localized heating at low temperatures. Due to sample variability, such as inter-layer CDW phase differences and mechanical variability such as tears or holes in flakes, these limits are approximate in general scanning. In cases where samples transformed into bulk prismatic phase, they are better observed.

**Modeling the CDW twinning across the C ⇌ IC transition.** In general, a CDW is characterized by two intertwined ingredients: (1) the amplitude/phase of the CDW order parameter and (2) the length/orientation of the CDW wavevector. The significant role of the first ingredient has been extensively studied in literature[21,34,56,57]. For this study, we focus on the wavevector orientation ($\theta$) and its fluctuation as a minimum model to characterize the competition between the two mirror twins observed in the experiments. This approximation correctly captures qualitative features away from the IC-C CDW transition. Near the phase transition, where additional fluctuations are no longer negligible, a more sophisticated model becomes needed.

We approximate the free energy landscape using Landau expansion:

$$f_i(T) = a_2(T - T_c)\theta_i^2 + a_4\theta_i^4 + \sum_{nn}^{6} \cos(\theta_{nn} - \theta_i)^2, \quad (1)$$

with $f$, $T$, $T_c$, $a$, $\theta$ denotes local free energy, temperature, transition temperature, Landau energy coefficient, and local CDW orientation respectively. The last term is an XY nearest-neighbor interaction that enforces smoothness; in the continuum limit, it converges to $|\nabla\theta|^2$. We chose six nearest-neighbors to accommodate the crystal symmetry (Fig. 3c). The simulation was done on hexagonal grid with 65,536 sites and periodic boundary condition.

The distribution of $\theta$ was calculated using Markov Chain Monte Carlo simulation with Metropolis-Hastings algorithm. Initially, a random distribution $\theta$ was generated. At each iteration, 40% of sites were randomly selected, then randomly generated $\theta$ were accepted or rejected based on Boltzmann statistics: $\exp[-\Delta f/k_B T]$. The effect of cooling was simulated in simulated annealing manner where initial $T$ was set to $2T_c$ then reduced by $0.2T_c$ every $10^{10}$ iterations (See Supplementary Fig. S9 for simulation parameters).

To simulate far-field diffraction from simulated $\theta$, each lattice position ($\mathbf{r}_i$) was distorted with three longitudinal modulation waves with wave vector $\mathbf{q}_{i,1}$, $\mathbf{q}_{i,2}$, $\mathbf{q}_{i,3}$ along $\theta_i$, $\theta_i + 120°$, $\theta_i + 240°$: $\mathbf{r}' = \mathbf{r}_i + \sum_{n=1,2,3} \mathbf{A}_n \sin(\mathbf{q}_{i,n} \cdot \mathbf{r}_i)$. Far-field diffraction ($I(\mathbf{k})$) was calculated by taking modulus squared of plane waves from each lattice sites: $I(\mathbf{k}) = |\sum_i \exp[i\mathbf{k} \cdot \mathbf{r}']|^2$.

**RA-SHG measurements.** The RA-SHG measurements were performed with the beam at normal incidence. The reflected SHG intensity is recorded as a function of the azimuthal angle between the incident electric polarization and the in-plane crystalline $a$-axis, with the reflected electric polarization being parallel to the incident one. In this experiment, the incident ultrafast light source was of 800 nm wavelength, 50 fs pulse duration and 200 kHz repetition rate, and was focused with a 5 μm diameter spot on the sample with a fluence of ~0.2 mJ/cm$^2$. The intensity of the reflected SHG was measured with a single photon counting detector. SHG measurements was performed on TaS$_x$Se$_{2−x}$ sample, which has same point group symmetries with TaS$_2$ both before and after heat treatment.

**Absorption measurements.** A home-built microscopic system with tungsten halogen source (Ocean Optics HL-2000-LL0) was used for absorption measurement. The transmitted signal was directed to Princeton Instruments IsoPlane-320 spectrometer, dispersed by 1200 grooves per mm diffraction grating, and detected by PIXIS: 400BR CCD camera.

## Data availability
All data and algorithms supporting the findings of this paper are described in "Methods" section and Supplementary materials. Additional data is available from the corresponding author upon reasonable request.

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

## Acknowledgements

S.H.S. acknowledges the financial support of the W.M. Keck Foundation. L.Z. acknowledges support by NSF CAREER grant No. DMR-174774 and AFOSR YIP grant No. FA9550-21-1-0065. Experiments were conducted using the Michigan Center for Materials Characterization (MC2) with assistance from Haiping Sun and Bobby Kerns and the Platform for the Accelerated Realization, Analysis, and Discovery of Interface Materials (PARADIM) supported by the National Science Foundation under Cooperative Agreement No. DMR-2039380. This work was performed in part at the University of Michigan Lurie Nanofabrication Facility with assistance from Pilar Herrera-Fierro, Vishva Ray, and Sandrine Martin—supported by the National Science Foundation Major Research Instrumentation award under No. DMR-1428226. Y.L., W.J.L., and Y.P.S. thank the support of the National Key Research and Development Program under contracts 2016YFA0300404 and the National Nature Science Foundation of China under contracts 11674326, 11774351. T.H.B. acknowledges support from the Office of Naval Research through the Base Program at the Naval Research Laboratory. P.B.D. acknowledges support of the Army Research Office grant No. W911NF2010196. We thank Lu Li for helpful discussion.

## Author contributions

S.H.S., N.S., and R.H. performed HAADF-STEM and in-situ TEM. S.H.S., N.S., R.H., I.B., and L.F.K. performed 4D-STEM experiments. S.H.S., S.N., J.G., T.H.B., and J.T.H. fabricated samples for electronic measurements. S.N., N.M.V., and J.T.H. performed electronic measurements. X.L. and L.Z. performed RA-SHG measurements. P.B.D. and Z.L measured optical absorption. Y.L., W.J.L., and Y.S. grew 1T-TaS$_x$Se$_{2−x}$ crystals. S.H.S., R.H., and K.S. provided theoretical interpretation. S.H.S. and R.H. prepared the manuscript. All authors reviewed and edited the manuscript.

## Competing interests

The authors declare no competing interests.
