## [Peer Review File · Nature Communications]

REVIEWER COMMENTS

Reviewer #1 (Remarks to the Author):

Sung et al.

The paper is reasonably well written and very nicely presented. It shows some very impressive results. I think they clearly demonstrate an interesting phase transition. I think the paper is of broad interest and well suited to this journal. However, I thought the abstract was a little bit over-hyped, so I think it can be accepted, but I do suggest some revisions before publication.

It needs a few English corrections. Mostly I don't think these obscure the meaning; primarily it is missing a few articles and so on. However, that should be fixed.

I've got quite a lot of thoughts, but as this can all be easily addressed, I'll list them here:

One thing that did confuse me is the "clean limit". Normally I would associate this term with the mean free path compared to the coherence length, at least for superconductors. However, I think the authors use this term here, without really getting into what they mean, but they also use "clean" in the housecleaning sense. I'm not clear precisely how they established either the coherence or free path for their sample. Is this in-plane or out-of-plane coherence and so on? As this term is used a lot in the paper, I would like to see a bit of discussion and clarification.

Perhaps some statistics on the grain size could help clarify this point. That might also be relevant for fig 2a and S3 – how does the grain size compare to the area for the diffraction patterns? How thick was the region? Also for fig 2b-e how thick is this area?

Same for Fig 1: How thick is the sample?

Perhaps my only remaining scientific doubt is whether there could be some other phase (or mixture of phases). Their verification of the CDW is likely to be correct, but all the evidence is a little indirect. This figure and the high spatial resolution are quite convincing. Presumably in thinner regions they also find areas with only one CDW direction?

It seems that the authors have convincingly demonstrated a phase change controlled by their heat treatment. I am not quite sure that they have completely sealed the case that they are seeing the charge density wave. For example, could a mix of 1T with other phases give similar patterns? I think I would like the logic spelled out a bit more clearly. I would really like them to work through what the diffraction patterns from the mix of phases they show (figure 2 or supplemental figure 6) would give both with and without a CDW and show how that reinforces their result. The diffraction data and

simulation in the supplemental almost does this. Perhaps labelling fig S4 and a few extra simulations in the supplemental could really spell this point out. What exactly are the phases in the model in fig S8? (That figure is almost what I'm asking for, but there are quite a lot of phases and mixtures that don't seem to be included. Although the caption is a bit brief, so it's hard to be sure.)

I would like to see the diffraction patterns of all the structures in Fig S1. I think most of my concerns here can be addressed if they show in the supplemental that no mixture of these phases could be confused with the CDW in diffraction.

Also S1 could perhaps be a bit more instructive. Labelling axes and adding color to indicate bonding might be considered.

I think some of the claims are a little stretched. For example, “engineered” appears in the conclusions, but without any real explanation. Fig S5b is impressive, but to claim “engineering” I would like to see a bit more control over this process and some evidence for how repeatable it is. For example, statistics on the size of the regions and number of layers and reproducibility would help. Either that, or perhaps soften this language a little. I might say that they “have established a pathway towards...”

On a similar note, the “reducing their dimension to 2d” and similar phrases are perhaps a bit over-reaching. The materials are layered, and so essentially already two-dimensional.

What I think they've done is show that heat treatment seems to give a layer-by layer phase change in an exciting material. Mapping this change is interesting and new. It probably supports some really interesting quantum properties, although I think to talk about the stabilizing quantum phases might need them to demonstrate measurement of such effects.

For example, their reference 34 uses EELS to show the existence of a CDW. If the current authors want to prove the 2D CDW stabilization, due to metallic separating layers, they might be able to demonstrate that with an EELS scan.

The title is reasonable, but for my taste the abstract is a little too overblown. I didn't like the “only in unrealized clean limit 2D materials such as 1T-TaS₂” – I'm not sure the material is “unrealized”, nor that it the only such candidate material. Also I think “engineering” is not an accurate description for heat treatment, unless they demonstrated, say, really precise control of the number and type of layers.

Page 1. “This stabilizes a collective insulating ground state not expected to exist at room temperature.” – I think I was a bit confused by which phase is meant.

Fig 1d. The diffraction pattern looks a bit strange. Possibly the conversion to pdf didn't work. Please add a high-res version to the supplemental.

Page 2: Why write 10^2 nm/s instead of 100 nm/s?

There are a few sentences that are not really clear, for example on Page 4: “Both structural and electronic investigations show the disordered NC-CDW phase disappears when a 2D CDW in a chemically and epitaxially native environment.”

Fig S5: Please also give the right hand image without the color overlay. I think that's important, because the S atoms are not all that well resolved. It's hard to be sure if there is some intermediate phase.

Reviewer #2 (Remarks to the Author):

The Authors describe spatially-resolved electron diffraction and TEM data on heat-treated TaS₂ flakes. They show that their heat treatment induces a layer-by-layer structural transition from the 1T (or 'Oc') polytype toward the 2H (or 'Pr') polytype. They also show that on subsequent cooling, the sample does not enter the NC-CDW phase which is usual for 1T-TaS₂, but instead features residual isolated 1T layers amid the bulk which is largely metallic Pr structure, as seen in the TEM images. The isolated 1T layers are shown to host the 'star-of-David' C-CDW at an enhanced temperature of ~350 K. These 1T layers are decoupled such that either one of the mirror symmetry partners of the C-CDW pattern may be instantiated in any given 1T layer, which matches the outcome of the Authors' Monte Carlo simulations. The Authors suggest these isolated C-CDW layers could host interesting but fragile quantum phases such as a spin liquid.

The work is clearly novel and important for the field of 2D correlated systems. The paper is exceptionally well-presented and accessible, and mostly well-written. Although I am not an expert in all the techniques used, it is clear that the work is technically very impressive, and the subsequent arguments made by the Authors are convincing. I appreciate the thorough and informative description of the sample preparation, treatments, and subsequent shelf-life. This is a valuable and timely addition to the methods for accessing truly 2D correlated behavior in TaS₂.

The only significant criticism I have is that a few of the citations in the introduction do not really seem cogent to the material under discussion: Are references 3, 6 and 7 really applicable to the metal-insulator transitions and superconductivity appearing in TaS₂?

Nevertheless, as long as the Authors make sure all the citations are all suitably used, for the reasons given above I think the manuscript should be accepted almost as it is.

PS. I have minor suggestions that the Authors can also consider:

i) A linguistic point -- Instead of 'interdigitated', perhaps 'interleaved' would be a better description of the arrangement of Oc and Pr layers. It emphasizes the stacked-2D nature of the system, whereas 'interdigitated' seems more apt for only 0D or 1D objects.

ii) There is a typo -- In the Methods section, 'Metropolis-Hasting' should be 'Metropolis-Hastings'.

Reviewer #3 (Remarks to the Author):

In this manuscript, the authors report the discovery of the out-of-plane tC-CDW phase. The authors start with a 1T-TaS₂ single crystal and heat it up to 720K such that the 1T polytype transforms to the 2H polytype layer-by-layer. From diffraction measurement, electric transport, and optical measurement, the authors suggest that α -CCDW or β -CDW phase are obtained in the remaining 1T layers, forming an out-of-plane tC-CDW phase. The authors claim that this phase appears because the metallic 2H layers screen interaction between 1T layers. We highly appreciate the authors' experiments. However, their analysis, interpretation, and theory have many speculations and some arguments are incorrect. We will reconsider whether the manuscript can be published after the following issues are clarified.

Major Comment 1:

In the introduction, the authors write that they discovered the out-of-plane tC-CDW without explaining what it is. What the authors discovered should be clarified in the introduction. In the result section the tC-CDW phase is explained as:

"The tC-CDW phase reported herein has distinct out-of-plane charge order—illustrated in Figure 1a. 2D CDWs reside within Oc-layers sparsely interdigitated between metallic Pr-layers. Each CDW is commensurate in one of two degenerate twin states, α -C (blue) or β -C (red) (Fig. 1b, c)."

and

"Both α and β CDW states were mapped over microns of area to reveal a uniform co-existence of both twins when viewed in projection out-of-plane (Fig. 1e,f). This is evidently different from recently reported in-plane twin CDWs created by femtosecond light pulses [31]."

However, we have the following questions about the nature of the tC-CDW phase.

1. It is not clear what Fig. 1 (e,f) shows. Is it a dark field image? Diffraction images usually give information of several layers. So, the sample thickness and how many layers appear in the diffraction image must be clarified. Moreover, the difference from Ref[31] must be explained quantitatively.
2. Layers with α -CCDW only or β -CCDW have not been observed so far. How can the authors confirm that such states appear in their results? In principle, whether the authors get an in-plane α/β multidomain or out-of-plane α/β cannot be determined from diffraction measurement. Such determination is possible only with layer-by-layer measurement, for instance, using STM.
3. The authors argue that metallic layers screen inter-layer interaction. This speculation needs further support. It should be investigated layer-by-layer with photoemission spectroscopy, STS, etc.
4. Wilson's multidomain boundary (Ref. [1]) is also possible. The authors argue that the restoration of mirror symmetry is important, but mirror symmetry is also preserved for in-plane α/β multidomain.
5. Fig. 4a: The 13x13 structure is shown without explanation. Its relation to the tC-CDW is not clear.

Major Comment 2:

It is not clear whether the authors' result can be distinguished from α/β superposition in 4Hb-TaS₂.

1. The authors explain that the change in polytype structure can be seen from the contrast in TEM image, but whether 1T, 2H, or 4Hb is obtained cannot be determined. Essentially, how is the polytype structure distinguishable from 4Hb? Moreover, 4Hb may also appear in Fig2i.
2. In Fig. S6 the authors write "This is evidently distinct from 4Hb-TaS₂ -- an ordered bulk crystal phase with alternating Pr-Oc stacking" but this difference must be explained quantitatively.
3. Resistance measurement of heat-treated sample in Fig4a resembles the typical resistance of 2H or 4Hb in Ref [1]. How can the temperature dependence of resistance be distinguished from 4Hb?

Therefore, the authors must analyze pristine 4Hb-TaS₂, show results like Fig.1 (e,f), compare the result with their polytype structure, and clearly explain the difference in comparison with Ref [1].

Major Comment 3:

The authors show results for TaS₂ polytypes in the whole manuscript except in Fig. 2 (h,i). In the caption of Fig. 2 (h,i) the authors write "A selenium doped sample was imaged to enhance chalcogen visibility" but the authors must explain clearly if they obtain a similar image without Se doping. S and Se have different physical properties, so the value of x for Se doping must be specified. 2H-TaS₂ which has different CDW phases and different superlattice vectors can also appear. It is not appropriate to interpret their result based on a different material.

Major Comment 4:

The authors completely ignore 45 years of CDW free energy theory in transition metal dichalcogenides, which is very well-understood. CDW is a macroscopic quantum condensate, so free energies like in Ref [44] must be used. The CDW free energy of TaS₂ is very well understood [JPSJ 43, 1839 (1977); JPSJ 44, 1465 (1977)]. The authors also discuss clean monolayer, but free energy in monolayer TaS₂ is also well understood [Sci. Rep. 10, 1239 (2020)].

Moreover, Eq. 1 leads to many questions as follows:

- The authors do not discuss CDW amplitudes like in the free energy theories cited above.

- The validity of Eq. 1 is not clear. Can it be derived from free energies with a macroscopic order parameter, or is it derived somewhere?
- The authors do not discuss discommensuration. Many effects like commensurability energy, lock-in energy, or triple-Q stabilization are ignored.
- It is not clear what the hexagonal grid in Fig3 represents. Does it represent a single David star unit, several CCDWs, or α/β domain?
- In the caption of Fig. 3 (f) the authors write "At high-temperature θ is mean-centered and disordered": does this mean the incommensurate phase? In this case, the authors are wrong. The IC phase in TaS₂ is not a collection of C-CDWs with random phases. Diffuse peaks do not appear for the IC phase in clean TaS₂ single crystals, and the IC phase is described by a single superlattice vector and a single phase.
- The IC-C phase transition is a first-order phase transition, so the inset in Fig. 3, which shows a second-order phase transition, is wrong. The IC and NC phase must also appear in the free energy.

Major Comment 5

The authors write that "Metallic layers screen impurity potentials to suppress the disordered nearly-commensurate (NCCDW) phase." However, impurity or disorder has nothing to do with the NC phase. It is well understood from experiments and theories (previous comment) that interference between CDW harmonics is essential for the appearance of the NC phase. The authors must consider the effect of CDW harmonics in their free energy analysis. Otherwise, whether the tC-CDW phase is more stable than the NC phase (or other phases) is ignored intentionally, which cannot be accepted.

Therefore, why the NC phase was not observed must be clarified based on the previous understanding of 1T-TaS₂. Note also that one cannot distinguish between C and NC by resistance measurement, and diffraction images must be considered.

Minor Comments

1. In the introduction the authors write "1T polytype ... exhibits three distinct CDW phases: commensurate (C), nearly-commensurate (NC), and incommensurate (IC)" but this is wrong. There is

also the triclinic (quasi-stripe) phase in 1T-TaS₂: See X-ray experiment [Synthetic Metals 11, 85 (1985)], STM experiment [Adv. Phys 37, 559 (1988)], and recent development such as [Sci. Rep. 10, 1239 (2020)], [Sci. Rep. 9, 7066 (2019)], and [npj Quantum Mater. 4, 32 (2019)]. There is also the stripe phase in 2H-TaSe₂. These phases cannot be ignored, and they must be discussed, including why they do not appear in Fig. 4 a.

2. The C-CDW transition temperature increases drastically when the sample thickness decreases to a few layers. For instance, the C-CDW phase is obtained in tri-layer 1T-TaS₂ above room temperature (Ref. [38]). For this reason, the authors write "free-standing ultrathin 1T-TaS₂ degrades long-range order [38]" but this is wrong at least for three layers. Therefore, the authors should discuss the increase of transition temperature in comparison with these facts.

3. We get a strong impression from Fig. 1 (a) that α/β appear alternately, which is highly unlikely to occur.

4. In Fig. 1 (d), the raw diffraction image of tC-CDW must be shown first without red and blue points (is it shown in FIG. S3?)

5. The authors write "Here, each 2D 1T-TaS₂ CDW is in its native chemical, epitaxial, and unstrained environment." Does "epitaxial" mean molecular beam epitaxy or something else?

6. In the "Synthesis of 1T-TaS_xSe_{2-x} crystal" section in Methods, the temperature unit is not shown correctly.

Therefore, we will reconsider whether the manuscript can be published after the above issues are clarified. The manuscript needs a major reconsideration based on these comments. Unfortunately, it must be said that the basic understanding of CDW in 1T-TaS₂ is lacking.

Robert Hovden, PhD
Assistant Professor
Materials Science & Eng.
University of Michigan
hovden@umich.edu
hovdenlab.com
c. 770-265-4042

Response to reviewers

Reviewer #1:

The paper is reasonably well written and very nicely presented. It shows some very impressive results. I think they clearly demonstrate an interesting phase transition. I think the paper is of broad interest and well suited to this journal.

We appreciate positive comments from the reviewer.

However, I thought the abstract was a little bit over-hyped, so I think it can be accepted, but I do suggest some revisions before publication.

To mitigate the hype in the abstract, we have removed “clean-limit” and “Engineered” has been replaced with “have established a pathway towards...” so that the observation of an interesting phase transition better stands on its own merit.

It needs a few English corrections. Mostly I don't think these obscure the meaning; primarily it is missing a few articles and so on. However, that should be fixed.

I've got quite a lot of thoughts, but as this can all be easily addressed, I'll list them here:

One thing that did confuse me is the “clean limit”. Normally I would associate this term with the mean free path compared to the coherence length, at least for superconductors. However, I think the authors use this term here, without really getting into what they mean, but they also use “clean” in the housecleaning sense. I'm not clear precisely how they established either the coherence or free path for their sample. Is this in-plane or out-of-plane coherence and so on? As this term is used a lot in the paper, I would like to see a bit of discussion and clarification.

We have modified all instances of “clean-limit” to read “clean”:

Page 1: Abstract “... exist only in unrealized clean 2D materials”

Page 1: Introduction, First paragraph “... and clean 2D charge density waves of superconductivity ...”

Page 1: Introduction, First paragraph “... and clean 2D confinement could enable a paradigm shift ...”

Page 2: Section “Clean-limit Polytype Heterostructures” is now “Clean Polytype Heterostructures”

Page 5, the first sentence of “Discussion” section: we have softened our language to “In summary, we have established a pathway toward stabilizing latent CDWs with long-range order at room temperature by reducing their dimensions to 2D using interleaved polytypes of TaS₂.”

Perhaps some statistics on the grain size could help clarify this point. That might also be relevant for fig 2a and S3 – how does the grain size compare to the area for the diffraction patterns?

Robert Hovden, PhD
Assistant Professor
Materials Science & Eng.
University of Michigan
hovden@umich.edu
hovdenlab.com
c. 770-265-4042

Grain sizes far exceed the $\sim(500 \text{ nm})^2$ areas measured by diffraction. SAED patterns on Fig. 2a and S5 were formed from 500 nm selected area aperture to grain sizes. To clarify this, Figure 2a caption on page 3 now reads "... SAED patterns were from a 500 nm aperture". Fig. S5 on page 6 of Supplemental Information now reads "In-situ TEM SAED Patterns: Raw data for Fig. 2a)."

*How thick was the region? Also for fig 2b-e how thick is this area?
Same for Fig 1: How thick is the sample?*

On page 5, we have added a new method subsection "Preparation of TEM samples" to clarify sample thickness information as well as preparation process:

Preparation of TEM samples

Both samples for 4D-STEM and TEM were prepared by exfoliating bulk 1T-TaS₂ and 1T-TaS_xSe_{2-x} crystals onto to polydimethylsiloxane (PDMS) gel stamp. The sample was then transferred to TEM grids using home-built transfer stage. Silicon nitride membrane window TEM grid with 2 μm holes from Norcada and DENS solution Wildfire Nano-chip grid were used. *From optical contrast and CBED patterns, the samples (Fig. 1, 2) were estimated to be 20 – 50 nm thick [Li_2013, Hovden_2018].*

[Li_2013]: Li, H et al. *ACS Nano* **7**, 10344-10353 (2013)

[Hovden_2018]: Hovden, R et al., *Microsc. Microanal.* **24**, 387-395 (2018)

Also, on page 5 Method subsection "Electron Microscopy", the last sentence of paragraph 2 "Both sample for 4D-STEM and TEM were prepared by exfoliating bulk 1T-TaS₂ crystal on to PDMS gel stamp, then transferring on to a TEM grid." was deleted to remove duplicate information.

Perhaps my only remaining scientific doubt is whether there could be some other phase (or mixture of phases). Their verification of the CDW is likely to be correct, but all the evidence is a little indirect. This figure and the high spatial resolution are quite convincing. Presumably in thinner regions they also find areas with only one CDW direction?

As suggested, we have now added new experimental data with fewer CDW layers (effectively thinner) to confirm emergence of one stronger CDW direction can occur. On page 3, 2nd paragraph of "Isolation of 2D Charge Density Waves" section, we have added "...In systems with very few CDW layers, one stronger CDW direction can sometimes emerge (Fig. S12b, c) due to an uneven ratio of α and β layers." On page 13 of Supplemental Information, we have added Figure S12 which shows thermally treated TaS₂ in tC-CDW, α C-CDW and β C-CDW phase at room temperature. These figures show that heating sample above 85°C to incommensurate (IC) phase then cooling down can result either tC-CDW or single-domain commensurate (C) CDW. This is possible with fewer octahedral / CDW layers (equivalent to a thin sample), it is more likely to have a single or stronger C-CDW direction.

Robert Hovden, PhD
Assistant Professor
Materials Science & Eng.
University of Michigan
hovden@umich.edu
hovdenlab.com
c. 770-265-4042

It seems that the authors have convincingly demonstrated a phase change controlled by their heat treatment. I am not quite sure that they have completely sealed the case that they are seeing the charge density wave. For example, could a mix of 1T with other phases give similar patterns? I think I would like the logic spelled out a bit more clearly. I would really like them to work through what the diffraction patterns from the mix of phases they show (figure 2 or supplemental figure 6) would give both with and without a CDW and show how that reinforces their result. The diffraction data and simulation in the supplemental almost does this. Perhaps labelling fig S4 and a few extra simulations in the supplemental could really spell this point out.

We appreciate the reviewer's question regarding CDW diffraction, which can be addressed two-fold: The most obvious smoking gun here is that we can switch tC-CDW to IC-CDW by heating above 80°C. If tC-CDW superlattice peaks (Fig. 1d and 2a) were due to mix of polytypes without CDW, the superlattice peaks should remain constant when heated.

In addition, stacking order and polytype manifests only as change in Bragg peak intensities [Sung_2018]. Furthermore, the intensity distribution of superlattice peaks follow the typical characteristic of CDW/PLD superlattice peaks. As shown by Overhauser [Overhauser_1975], as well as in [Wilson_1975] and [Hovden_2016], the relative intensity of CDW/PLD superlattice peaks to Bragg peaks increase as function of distance to the k-space origin. This is clearly distinguished from superlattice peaks from chemical ordering where the peak decreases in intensity. We have added the following discussions to clarify this point.

Page 2: The second paragraph of section "Twinned Commensurate Charge Density Waves" now reads "...Polytypes and stacking order thereof manifests as changes in Bragg peak intensities whereas CDWs produce distinct superlattice peaks. [Overhauser_1971, Hovden_2016, Sung_2018] (Supplemental section SI3)"

Page 4 of SI, 2nd Paragraph of SI 3: ... "For a typical value of \mathbf{A} , $J_{nn}(\mathbf{k}\cdot\mathbf{A})$ monotonically increases with \mathbf{k} . Therefore, CDW superlattice peaks become brighter relative to Bragg peaks near higher order Bragg peaks.

[Overhauser_1971]: Overhauser, A. W., *Phys. Rev. B* **3**, 3173-3182 (1971)

[Wilson_1975]: Wilson, J. et al., *Adv. Phys.* **24**, 117-201 (1975)

[Hovden_2016]: Hovden, R. et al., *PNAS* **113**, 11420-11424 (2016)

[Sung_2018]: Sung et al., *Phys. Rev. Mat.* **3**, 064003 (2018)

What exactly are the phases in the model in fig S8? (That figure is almost what I'm asking for, but there are quite a lot of phases and mixtures that don't seem to be included. Although the caption is a bit brief, so it's hard to be sure.)

On page 11 of Supplemental Information, Fig. S8 (now Fig. S10) is now annotated to highlight NC and C superlattice peaks. The caption on Fig. S10 now reads "a) Room temperature SAED, taken after partial polytypic transitions have occurred, shows both α -NC and β -C CDW peaks where many adjacent (i. e. coupled) CDW layers remain alongside isolated 2D CDW layers, resulting in the presence of both NC and C CDW respectively. Superlattice reflections for NC and C phases are marked cyan and magenta."

Robert Hovden, PhD
Assistant Professor
Materials Science & Eng.
University of Michigan
hovden@umich.edu
hovdenlab.com
c. 770-265-4042

In addition, Fig S5 of SI now includes schematic diagram of superlattice peak structures for NC, IC and tC CDW states for further clarification of diffraction patterns.

I would like to see the diffraction patterns of all the structures in Fig S1. I think most of my concerns here can be addressed if they show in the supplemental that no mixture of these phases could be confused with the CDW in diffraction.

On page 3 of SI, we have added a new SI (Fig. S3) with simulated electron diffraction patterns for 1T, 2Ha, 3R, 4Hb and 6R. As discussed in a previous response, polytypes and stacking order thereof manifest as change in Bragg peak intensities whereas CDWs produce distinct superlattice peaks. Fig. S3 caption reads “**Diffraction of TaS₂ Polytypes a–e**) Quantum mechanical multislice simulations of SAED for a) 1T, b) 2Ha, c) 3R, d) 4Hb and e) 6R TaS₂ without CDWs. Polytypes and stacking order thereof manifests as distribution of intensities at Bragg peaks. For example, first order Bragg peaks are forbidden reflections for 3R and 6R polytypes due to out-of-plane arrangement. Each simulation was computed for 12 van der Waals layers, with 30 frozen phonon configurations”

Also S1 could perhaps be a bit more instructive. Labelling axes and adding color to indicate bonding might be considered.

Page 3 of SI: Axes labels and bonding indicators are now marked on Fig. S1 (Now Fig. S2)

I think some of the claims are a little stretched. For example, “engineered” appears in the conclusions, but without any real explanation. Fig S5b is impressive, but to claim “engineering” I would like to see a bit more control over this process and some evidence for how repeatable it is. For example, statistics on the size of the regions and number of layers and reproducibility would help. Either that, or perhaps soften this language a little. I might say that they “have established a pathway towards...”

“Engineered” has been removed and replaced with “have established a pathway towards...” on page 5 in the first sentence of “Discussion” section. The softened language now reads “In summary, we have established a pathway toward stabilizing latent CDWs with long-range order at room temperature by reducing their dimensions to 2D using clean interleaved polytypes of TaS₂.”

On a similar note, the “reducing their dimension to 2d” and similar phrases are perhaps a bit over-reaching. The materials are layered, and so essentially already two-dimensional.

We appreciate the reviewer for a very helpful comment from the referee. Although intralayer couplings dominate energy scales, interlayer couplings in this material are non-negligible, making it quasi-2D. This is evident in bulk 1T-TaS₂ where NC and C CDW choose a single domain, breaking twin degeneracy. It is necessary to suppress interlayer coupling in order to observe certain 2D characteristics. Similar phenomena have been studied in other quasi-2D materials, such as cuprate high T_c superconductors [Berg_2007], where the ideal 2D limit is achieved through dynamical layer decoupling. In our study, we provided a different pathway to achieve a similar decoupling, via polytype heterostructures, to observe 2D characteristics.

Robert Hovden, PhD
Assistant Professor
Materials Science & Eng.
University of Michigan
hovden@umich.edu
hovdenlab.com
c. 770-265-4042

[Berg_2007]: Berg, E et al., *Phys. Rev. Lett* **99**, 127003 (2007)

What I think they've done is show that heat treatment seems to give a layer-by-layer phase change in an exciting material. Mapping this change is interesting and new. It probably supports some really interesting quantum properties, although I think to talk about the stabilizing quantum phases might need them to demonstrate measurement of such effects.

For example, their reference 34 uses EELS to show the existence of a CDW. If the current authors want to prove the 2D CDW stabilization, due to metallic separating layers, they might be able to demonstrate that with an EELS scan.

We appreciate the positive remarks regarding our report of the layer-by-layer phase change in an exciting material.

Qiao et al. [Qiao_2017] is impressive work that distinguishes very subtle change in Titanium EELS fine structure due to CDWs. However, as elaborated above (and now discussed in Section 3 of SI), the superlattice peaks are the clearest validation of periodic lattice distortion (PLD) from CDW. Additionally, Fig. 2a and Fig. S5 shows superlattice CDW peaks changes at high temperature. This is another telltale sign that the superlattice peaks are from CDW.

[Qiao_2017]: Qiao et al., *Phys. Rev. Mat.* **1**, 054002 (2017)

The title is reasonable, but for my taste the abstract is a little too overblown. I didn't like the "only in unrealized clean limit 2D materials such as 1T-TaS₂" – I'm not sure the material is "unrealized", nor that it the only such candidate material. Also I think "engineering" is not an accurate description for heat treatment, unless they demonstrated, say, really precise control of the number and type of layers.

"Engineered" has been removed and replaced with "have established a pathway towards..." on page 5 in the first sentence of "Discussion" section. The softened language now reads "In summary, we have established a pathway toward stabilizing latent CDWs with long-range order at room temperature by reducing their dimensions to 2D using clean interleaved polytypes of TaS₂."

In [Yu_2015], Yu et al established that 1T-TaS₂ does not support CDW structure in low-dimensions. In [Tsen_2015] Tsen et al showed that dimensions could be further reduced with encapsulation, however, the CDWs degrade and the C phase disappears entirely. The results in this manuscript show that in a clean, epitaxial environment, 2D CDWs can be stabilized over a large area ($> 1\mu\text{m}^2$).

[Yu_2015]: Yu, Y. et al., *Nat. Nanotechnol.* **10**, 270-276 (2015)

[Tsen_2015]: Tsen, A. W. et al., *PNAS* **112**, 15054-15059 (2015)

Page 1. "This stabilizes a collective insulating ground state not expected to exist at room temperature." – I think I was a bit confused by which phase is meant.

To clarify, page 1, para 2 now reads "...This stabilizes a collective insulating ground state (i.e. C-CDW)".

Robert Hovden, PhD
Assistant Professor
Materials Science & Eng.
University of Michigan
hovden@umich.edu
hovdenlab.com
c. 770-265-4042

Fig 1d. The diffraction pattern looks a bit strange. Possibly the conversion to pdf didn't work. Please add a high-res version to the supplemental.

Page 13 of Supplemental Information now includes Fig. S12a which presents raw position averaged diffraction pattern without markings. Fig. 1d caption now reads "...; unmarked image available in Supplemental Figure S12a." The diffraction pattern is collected on a pixel-array detector (EMPAD) equipped with 128x128 pixels.

Page 2: Why write 10^2 nm/s instead of 100 nm/s?

Page 2, the second paragraph on "Clean-Limit Polytype Heterostructures" now reads "... Arrows indicate movement of Oc/Pr coordination boundary with fast-transition up to ~ 100 nm/s ..."

There are a few sentences that are not really clear, for example on Page 4: "Both structural and electronic investigations show the disordered NCCDW phase disappears when a 2D CDW in a chemically and epitaxially native environment."

On page 5 "Discussion" section, the sentence now reads "Both structural and electronic investigations show the disordered NC-CDW phase disappears when the CDW is reduced to 2D within a chemically and epitaxially native, clean environment."

Fig S5: Please also give the right hand image without the color overlay. I think that's important, because the S atoms are not all that well resolved. It's hard to be sure if there is some intermediate phase.

We agree providing non-annotated data is important. On page 8 of Supplemental Information, Figure S6 (now Fig. S7) now includes HAADF-STEM image without the color overlay.

Robert Hovden, PhD
Assistant Professor
Materials Science & Eng.
University of Michigan
hovden@umich.edu
hovdenlab.com
c. 770-265-4042

Reviewer #2:

The Authors describe spatially-resolved electron diffraction and TEM data on heat-treated TaS₂ flakes. They show that their heat treatment induces a layer-by-layer structural transition from the 1T (or 'Oc') polytype toward the 2H (or 'Pr') polytype. They also show that on subsequent cooling, the sample does not enter the NC-CDW phase which is usual for 1T-TaS₂, but instead features residual isolated 1T layers amid the bulk which is largely metallic Pr structure, as seen in the TEM images. The isolated 1T layers are shown to host the 'star-of-David' C-CDW at an enhanced temperature of ~350 K. These 1T layers are decoupled such that either one of the mirror symmetry partners of the C-CDW pattern may be instantiated in any given 1T layer, which matches the outcome of the Authors' Monte Carlo simulations. The Authors suggest these isolated C-CDW layers could host interesting but fragile quantum phases such as a spin liquid.

The work is clearly novel and important for the field of 2D correlated systems. The paper is exceptionally well-presented and accessible, and mostly well-written. Although I am not an expert in all the techniques used, it is clear that the work is technically very impressive, and the subsequent arguments made by the Authors are convincing. I appreciate the thorough and informative description of the sample preparation, treatments, and subsequent shelf-life. This is a valuable and timely addition to the methods for accessing truly 2D correlated behavior in TaS₂.

The only significant criticism I have is that a few of the citations in the introduction do not really seem cogent to the material under discussion: Are references 3, 6 and 7 really applicable to the metal-insulator transitions and superconductivity appearing in TaS₂?

We thank the reviewer for pointing this out. On Page 1, we have removed Ref. 3, 6 and 7, and replaced with more appropriate citations.

Ref 3: Replaced by Ref 1 [Wilson et al. (1975)]

Ref 6: R. Ang et al., "Atomistic origin of an ordered superstructure induced superconductivity in layered chalcogenides. *Nat. Commun.* **6**: 6091 (2015)

Ref 7: Li, L. et al., "Superconducting order from disorder 2H-TaSe_{2-x}S_x" *npj Quantum Mater.* **2**:11 (2017)

Robert Hovden, PhD
Assistant Professor
Materials Science & Eng.
University of Michigan
hovden@umich.edu
hovdenlab.com
c. 770-265-4042

Nevertheless, as long as the Authors make sure all the citations are all suitably used, for the reasons given above I think the manuscript should be accepted almost as it is.

PS. I have minor suggestions that the Authors can also consider:

i) A linguistic point -- Instead of 'interdigitated', perhaps 'interleaved' would be a better description of the arrangement of Oc and Pr layers. It emphasizes the stacked-2D nature of the system, whereas 'interdigitated' seems more apt for only 0D or 1D objects.

We appreciate the reviewer for the suggestion, and modified all instances of 'interdigitated' to 'interleaved':

Page 1, 2nd Paragraph of "Introduction" section: "... clean interleaved 2D polytypic heterostructures"

Page 1, 2nd Paragraph of "Introduction" section: "... interleaving disables interlayer coupling"

Page 2, 1st Paragraph: "...2D CDWs reside within Oc-layers sparsely interleaved between ..."

Page 2, 2nd Paragraph of "Clean Polytype Heterostructures" section: "...interleaved polytypic ..."

Page 2, last Paragraph: "... reveal the interleaved polytypic heterostructure..."

Page 3, 1st Paragraph: "...Interleaving isolates..."

Page 4, 2nd Paragraph: "... polytypic heterostructure with interleaved CDWs..."

Page 4, last Paragraph: "... The interleaved polytype herein stabilizes..."

Page 5, 1st Paragraph of "Discussion": "...2D using clean interleaved polytypes of TaS₂"

Page 6, "Thermal Synthesis of tC-CDW in TaS₂" section: "Interleaved 2D TaS₂...Once the interleaved polytype is fully established..."

Page 6 of SI, Fig. S5 caption: "...c) Heating above 720 K produces interleaved ..."

ii) There is a typo -- In the Methods section, 'Metropolis-Hasting' should be 'Metropolis-Hastings'.

Page 5, the second paragraph of "Modelling C <-> IC transition" section now reads "The distribution of θ was calculated using Markov Chain Monte Carlo simulation with Metropolis-Hastings algorithm."

Robert Hovden, PhD
Assistant Professor
Materials Science & Eng.
University of Michigan
hovden@umich.edu
hovdenlab.com
c. 770-265-4042

Reviewer #3:

In this manuscript, the authors report the discovery of the out-of-plane tC-CDW phase. The authors start with a 1T-TaS₂ single crystal and heat it up to 720K such that the 1T polytype transforms to the 2H polytype layer-by-layer. From diffraction measurement, electric transport, and optical measurement, the authors suggest that α -CCDW or β -CDW phase are obtained in the remaining 1T layers, forming an out-of-plane tC-CDW phase. The authors claim that this phase appears because the metallic 2H layers screen interaction between 1T layers. We highly appreciate the authors' experiments. However, their analysis, interpretation, and theory have many speculations and some arguments are incorrect. We will reconsider whether the manuscript can be published after the following issues are clarified.

Major Comment 1:

In the introduction, the authors write that they discovered the our-of-plane tC-CDW without explaining what it is. What the authors discovered should be clarified in the introduction. In the result section the tC-CDW phase is explained as:

"The tC-CDW phase reported herein has distinct out-of-plane charge order—illustrated in Figure 1a. 2D CDWs reside within Oc-layers sparsely interdigitated between metallic Pr-layers. Each CDW is commensurate in one of two degenerate twin states, α -C (blue) or β -C (red) (Fig. 1b, c)."

and

"Both α and β CDW states were mapped over microns of area to reveal a uniform co-existence of both twins when viewed in projection out-of-plane (Fig. 1e,f). This is evidently different from recently reported in-plane twin CDWs created by femtosecond light pulses [31]."

However, we have the following questions about the nature of the tC-CDW phase.

1. It is not clear what Fig. 1 (e,f) shows. Is it a dark field image? Diffraction images usually give information of several layers. So, the sample thickness and how many layers appear in the diffraction image must be clarified. Moreover, the difference from Ref[31] must be explained quantitatively.

The technique used in Fig. 1 (d,e,f) is 4D-STEM, where full convergent beam electron diffraction pattern is captured at each pixel position. Fig. 1d is averaged diffraction patterns from the entire field-of-view. Figure 1e and f were constructed by integrating signals from each set of superlattice peaks.

Zong et al. [Zong_2018] mapped CDW domains with a handful of selected area electron diffraction patterns (SAED). There are few key differences:

1. Our 4D-STEM data has about 4 nm resolution (close to one CDW wavelength), whereas each SAED pattern in [Zong_2018] came from about $(1\mu\text{m})^2$ area. Therefore, our experiment provides truly local information of CDW distribution.
2. [Zong_2018] mapped CDW domain by sparsely sampling 79 regions. Our experiment did 64x64 rastered scan (4096 regions).
3. [Zong_2018] is resulted from ultrafast optical excitation.
4. In [Zong_2018], CDWs are out-of-plane coupled in each domain. This agrees with our result where metallic layers decouple out-of-plane interaction, allowing out-of-plane twin.

Robert Hovden, PhD
Assistant Professor
Materials Science & Eng.
University of Michigan
hovden@umich.edu
hovdenlab.com
c. 770-265-4042

To clarify this, we have added more detailed introduction of this novel technique in Page 2, last paragraph of section “Twinned Commensurate Charge Density Waves.” The last paragraph now reads

“... Mapping CDW structure required a pixel array detector with 1,000,000:1 dynamic range [Tate_2016] allowing Bragg beams to be imaged while still maintaining single electron sensitivity near CDW peaks. A convergent beam electron diffraction (CBED; 0.55 mrad convergence semi-angle, 200 keV) pattern was collected at every beam position across large fields-of-view---an emerging technique often called 4D-STEM (See Methods) [Tate_2016, Panova_2019, Qiao_2017]. In this way, the local CDW structure was measured at ~4.6 nm resolution and across > 1 μ m fields of view. It demonstrates 4D-STEM as an invaluable tool for mapping charge order in materials...”

[Zong_2018]: Zong et al., *Sci. Adv.* **4**, eaau5501 (2018)

[Tate_2016]: Tate, M. W. et al. *Microsc. Microanal.* **22** 237-249 (2016)

[Panova_2019]: Panova, O. et al., *Nat. Mater.* **18**, 860-865 (2019)

[Qiao_2017]: Qiao, Q. et al., *Phys. Rev. Mat.* **1**, 054002 (2017)

2. Layers with α -CCDW only or β -CCDW have not been observed so far. How can the authors confirm that such states appear in their results? In principle, whether the authors get an in-plane α/β multidomain or out-of-plane α/β cannot be determined from diffraction measurement. Such determination is possible only with layer-by-layer measurement, for instance, using STM.

Using nanobeam diffraction is a way to overcome traditional limits of both real and reciprocal space measurements. Each diffraction pattern used to map CDW domain has a beam width of ~4.6 nm. It is very unlikely that in-plane α/β multidomains exist within such a small beam size. In addition, we’ve added Fig. S11 that shows complete conversion to single-domain C-CDW. This was obtained by heating the thermally-treated sample above 80 °C (T_{IC}) then cooling. Upon cooling, system randomly switches between α C-CDW, β C-CDW or α/β tC-CDW. Direct evidence that there are multiple decoupled layers in projection.

Furthermore, we have added new experimental data, as suggested by Reviewer #1, which shows in specimens with fewer CDW layers emergence of one CDW direction can occur. On page 11 of Supplemental Information, we have added Figure S12 which shows thermally treated TaS₂ in tC-CDW, α C-CDW and β C-CDW phase at room temperature. These figures show that heating sample above 85°C to incommensurate (IC) phase then cooling down can result either tC-CDW or single-domain commensurate (C) CDW. This is possible because each octahedral layer is going through IC->C transition independently. With fewer octahedral / CDW layers (equivalent to a thin sample), it is more likely to have a single or stronger C-CDW direction.

3. The authors argue that metallic layers screen inter-layer interaction. This speculation needs further support. It should be investigated layer-by-layer with photoemission spectroscopy, STS, etc.

We agree with the referee that combining the results from multiple experimental techniques would benefit our understanding about the new T_c enhancement mechanism observed in this work. On the other hand, it is also worthwhile to point out that the key result of our study is not to claim that the existence of metallic layers screening. Instead, the key focus is this unexpected connection between structure and T_c. Among all

Robert Hovden, PhD
Assistant Professor
Materials Science & Eng.
University of Michigan
hovden@umich.edu
hovdenlab.com
c. 770-265-4042

possible mechanisms, we find that metallic layers screening provides a natural explanation. Because this is the first study to report this new phenomenon, it is beyond the scope of our study to prove that this screening mechanism must be the sole and only origin of this new phenomenon. To fully conquer this problem, will require extensive studies and the combined efforts of multiple research groups with different experimental techniques, as the referee has pointed out. Therefore, we believe it is important to publish this work in a high-profile journal with a large audience, which will help generate interest and coherent efforts across multiple disciplines, and such efforts would eventually lead to a full understanding of this intriguing phenomenon.

4. Wilson's multidomain boundary (Ref. [1]) is also possible. The authors argue that the restoration of mirror symmetry is important, but mirror symmetry is also preserved for in-plane α/β multidomain.

In our work, we use a recently developed 4D STEM imaging technique (now further elaborated on page 2, para. 3) [Tate_2016, Panova_2019, Qiao_2017], that shows a homogenous distribution of α/β twins when viewed in projection, not in-plane domains. The in-plane domains, such as those created by Ref [1, 30] have measurably uneven spatial distributions, with reported domain sizes of many microns (note, domain sizes on Ref. [1] is ambiguous as the authors did not report exact scale for their film micrographs, but are likely to be multi-microns based magnification values).

[Tate_2016]: Tate, M. W. et al. *Microsc. Microanal.* **22** 237-249 (2016)

[Panova_2019]: Panova, O. et al., *Nat. Mater.* **18**, 860-865 (2019)

[Qiao_2017]: Qiao, Q. et al., *Phys. Rev. Mat.* **1**, 054002 (2017)

5. Fig. 4a: The 13x13 structure is shown without explanation. Its relation to the tC-CDW is not clear.

On page 5, Fig. 4a caption now reads "...Schematics represent simplified CDW structures of each phase ." for clarification. Colored outline was added to each schematic for more clarification as well.

Major Comment 2:

It is not clear whether the authors' result can be distinguished from α/β superposition in 4Hb-TaS2.

1. The authors explain that the change in polytype structure can be seen from the contrast in TEM image, but whether 1T, 2H, or 4Hb is obtained cannot be determined. Essentially, how is the polytype structure distinguishable from 4Hb? Moreover, 4Hb may also appear in Fig2i.

Cross sectional TEM diffraction (specifically the delocalized intensity between Bragg peaks Fig. S8b) and HAADF-STEM (the atomic coordination of each layer in Fig. 2i) clearly shows an interleaved polytype is obtained and is not the ordered 1T, 2H, or 4Hb polytypes. This work was reproduced on multiple samples.

Robert Hovden, PhD
Assistant Professor
Materials Science & Eng.
University of Michigan
hovden@umich.edu
hovdenlab.com
c. 770-265-4042

2. In Fig. S6 the authors write "This is evidently distinct from 4Hb-TaS₂ -- an ordered bulk crystal phase with alternating Pr-Oc stacking" but this difference must be explained quantitatively.

The obtained phase is clearly distinguished from 4Hb-TaS₂. The polytypic heterostructure obtained in this manuscript is clearly distinct as it is a random interleaving of Octahedral layers within Prismatic layers. This was shown in two different measurements.

1. HAADF-STEM micrographs in Fig. 2i and S7 shows random interleaving of octahedral layers within prismatic layers.
2. Cross-sectional SAED of thermally treated sample (Fig. S8b) shows streaking along [0001] which is absent in pristine 1T sample (Fig. S8a). This indicates disordered out-of-plane stacking. 4Hb polytype exhibits superlattice peaks along [0001] that corresponds to a 4-layer repeat, which we do not observe.

3. Resistance measurement of heat-treated sample in Fig4a resembles the typical resistance of 2H or 4Hb in Ref [1]. How can the temperature dependence of resistance be distinguished from 4Hb?

While it is true that the resistance measurement in Fig. 4a resembles that of 2H and 4Hb TaS₂, the figure shows distinctive behaviors.

1. 4Hb-TaS₂ features a vertical resistance jump at 85°C and very sudden change of slope at 20 K. Our polytypic heterostructure shows a mild slope at ~85°C and gradual change of slope at ~50 K.
2. 2H-TaS₂ resistance doesn't have a resistance jump at ~85°C, this is distinct IC->NC/C phase transition for octahedral layer.

Therefore, the authors must analyze pristine 4Hb-TaS₂, show results like Fig.1 (e,f), compare the result with their polytype structure, and clearly explain the difference in comparison with Ref [1].

As we have addressed above, we have demonstrated that our polytype heterostructure is evidently distinct from 4Hb-TaS₂.

Major Comment 3:

The authors show results for TaS₂ polytypes in the whole manuscript except in Fig. 2 (h,i). In the caption of Fig. 2 (h,i) the authors write "A selenium doped sample was imaged to enhance chalcogen visibility" but the authors must explain clearly if they obtain a similar image without Se doping. S and Se have different physical properties, so the value of x for Se doping must be specified. 2H-TaS₂ which has different CDW phases and different superlattice vectors can also appear. It is not appropriate to interpret their result based on a different material.

As the reviewer stated, all results including in-situ TEM, diffraction, conductivity, and SHG were done on TaS₂ and supports the interpretation of our results. CDW phases were not interpreted based on Selenium doped specimens. In-situ TEM of TaS₂ (Fig. 2b-e, Fig. S6, Supplemental Video) was used to directly observe the polytypic transition at the expected transition temperature. Both 1T-TaS₂ and 1T-TaSe₂ share the same structure and have highly comparable polytype transition at high temperatures [Wilson_1975].

Robert Hovden, PhD
Assistant Professor
Materials Science & Eng.
University of Michigan
hovden@umich.edu
hovdenlab.com
c. 770-265-4042

The same process was observed using $\text{TaS}_x\text{Se}_{2-x}$ which also provided a heavier element for atomic resolution HAADF imaging to additionally evidence the polytype transformation.

Value of x for Se doping has now been clarified on page 5 “Synthesis of 1T-TaS₂ crystal”. The section now reads “High-quality single crystals of 1T-TaS₂ and 1T-TaS_xSe_{2-x} ($x \approx 1$) ... All CDW characterization was done on 1T-TaS₂; Se-doped sample was used only for polytype characterization in cross-sectional HAADF-STEM (Fig. 2h, i, S7) and cross-sectional SAED (Fig. S8).”

Major Comment 4:

The authors completely ignore 45 years of CDW free energy theory in transition metal dichalcogenides, which is very well-understood. CDW is a macroscopic quantum condensate, so free energies like in Ref [44] must be used. The CDW free energy of TaS₂ is very well understood [JPSJ 43, 1839 (1977); JPSJ 44, 1465 (1977)]. The authors also discuss clean monolayer, but free energy in monolayer TaS₂ is also well understood [Sci. Rep. 10, 1239 (2020)].

Moreover, Eq. 1 leads to many questions as follows:

- The authors do not discuss CDW amplitudes like in the free energy theories cited above.*
- The validity of Eq. 1 is not clear. Can it be derived from free energies with a macroscopic order parameter, or is it derived somewhere?*
- The authors do not discuss discommensuration. Many effects like commensurability energy, lock-in energy, or triple-Q stabilization are ignored.*

We very much appreciate the references highlighted by the referees, which we have incorporated in the revised manuscript. We fully agree with the referee that as have been shown in existing literature, CDW order and fluctuations in CDW order involves many effects. In our theory analysis, instead of a full characterization that includes all these ingredients, which is a very challenging task and requires very careful analysis and modeling, our objective here is to demonstrate a minimum phenomenological model which incorporates a few key ingredients. Since two orientations are available and at first glance appear equivalent, we can model the energy as a double well potential. A recent study ([Zong_2018]) used a similar argument but in their case they added a strain term that lifts the degeneracy between the twins. Here, we show that kinetic means (quench) alone can lead to the formation of twins.

We now state on page 6, paragraph 3 of “Modelling C \leftrightarrow IC transition” section, “In general, a CDW is characterized by two intertwined ingredients: (1) the amplitude/phase of the CDW order parameter and (2) the length/orientation of the CDW wavevector. The significant role of the first ingredient has been extensively studied in literature [McMillan_1975, Nakanishi_1977, Nakanishi_1978, Nakatsugawa_2020]. For this study, the second ingredient is the key focus, which characterizes the fluctuation in the wavevector orientation θ as a minimum model to characterize the competition between the two phases observed in the experiments. This approximation correctly captures qualitative features away from the IC-C CDW transition. Near the phase transition, where additional fluctuations are no longer negligible, a more sophisticated model becomes needed.”

Robert Hovden, PhD
Assistant Professor
Materials Science & Eng.
University of Michigan
hovden@umich.edu
hovdenlab.com
c. 770-265-4042

Again the analysis does not aim to re-address the Landau expansion of the various CDW states for a single twin. We have revised the manuscript to highlight this point. Page 3, 1st paragraph of “Isolation of 2D Charge Density Waves” now reads “... This provides a simple, minimal phenomenological model to capture the formation of twinned CDWs (See Methods).” Full description of CDW requires more comprehensive order parameter and free energy expression.

[McMillan_1975]: McMillan W. L., Phys. Rev. B 12, 1187-1196 (1975)

[Zong_2018]: Zong et al., Sci. Adv. 4, eaau5501 (2018)

[Nakanishi_1977]: Nakanishi, K. and Shiba, H., J. Phys. Soc. Japan 43, 1839-1847 (1977)

[Nakanishi_1978]: Nakanishi, K. and Shiba, H., J. Phys. Soc. Japan 44, 1465-1473 (1978)

[Nakatsugawa_2020]: Nakatsugawa, K. et al., Sci. Rep. 10, 1239 (2020)

- It is not clear what the hexagonal grid in Fig3 represents. Does it represent a single David star unit, several CCDWs, or α/β domain?

We thank the reviewer for pointing this out. Each hexagonal cell represents a Ta ion site. On Page 4, Fig. 3, the diagram is now labeled as e, and Fig. 3e caption now reads “e) Map of local orientation of wave vector (θ) at IC phase. Each hexagonal cell represents θ at each Ta site” for clarification.

- In the caption of Fig. 3 (f) the authors write “At high-temperature θ is mean-centered and disordered”: does this mean the incommensurate phase? In this case, the authors are wrong. The IC phase in TaS₂ is not a collection of C-CDWs with random phases.

Here, θ refers to local orientation of CDW, not the phase of C-CDW. To reduce confusion, on page 4, caption for Fig. 3e now reads “...local orientation of wave vector (θ) ...” to clarify the definition of θ prior to Fig. 3f.

Diffuse peaks do not appear for the IC phase in clean TaS₂ single crystals, and the IC phase is described by a single superlattice vector and a single phase.

We respectfully disagree with the reviewer. A peak in reciprocal space represents the mean lattice vector and does not reveal phase information or local variation. The broad and diffuse superlattice peaks in the IC phase are associated with CDW disorder which increases as temperature is increased. [Vogelgesang_2017, Han_2015, ElBaggari_2018]

[Vogelgesang_2017]: Vogelgesang S. et al., Nat. Phys. 14, 184-190 (2017)

[Han_2015]: Han, T.-R. T. et al., Sci. Adv. 26, e1400173 (2015)

[ElBaggari_2018]: El Baggari, I. et al., PNAS 115, 1445-1450 (2018)

- The IC-C phase transition is a first-order phase transition, so the inset in Fig. 3, which shows a second-order phase transition, is wrong. The IC and NC phase must also appear in the free energy.

As discussed in a previous response, our model attempts to capture the formation of twins (the order parameter is the orientation and not the CDW order parameter) and neglects the triple-Q formation.

Robert Hovden, PhD
Assistant Professor
Materials Science & Eng.
University of Michigan
hovden@umich.edu
hovdenlab.com
c. 770-265-4042

Therefore, we do not expect a first-order transition. Even in the case of the full CDW Landau expansion, the IC-C transition is second order if only a single Q is modelled. Naturally, including the triple-Q state and the couplings (third-order couplings $A_1A_2A_3$) makes the transition first order.

Major Comment 5

The authors write that “Metallic layers screen impurity potentials to suppress the disordered nearly-commensurate (NCCDW) phase.” However, impurity or disorder has nothing to do with the NC phase. It is well understood from experiments and theories (previous comment) that interference between CDW harmonics is essential for the appearance of the NC phase.

As stated on page 1, 3rd paragraph, “At room temperature, the conductive NC-CDW is generally accepted as a C-CDW with short range order [Wu_1898, Vogelgesang_2017, Hovden_2016] that permits electron transport along regions of discommensuration [Cho_2016, Sipos_2008].” The aforementioned harmonics do reflect the presence of phase and amplitude disorder. To obtain the NC phase, the amplitude and phase of the CDW order parameter $A \exp[i\phi]$ are assumed to be spatially varying [Vogelgesang_2017]. An alternative way to see this is that the NC phase has phase slips between commensurate patches. Therefore, the harmonics are the other side of the coin of spatial phase variations (they are the expansion of phase slips).

[Wu_1898]: Wu, X. L. et al., *Science* **243**, 1703-1705 (1989)
[Vogelgesang_2017]: Vogelgesang S. et al., *Nat. Phys.* **14**, 184-190 (2017)
[Hovden_2016]: Hovden, R. et al., *PNAS* **113**, 11420-11424 (2016)
[Cho_2016]: Cho, D. et al., *Nat. Commun.* **7**, 10453 (2016)
[Sipos_2008]: Sipos, B. et al., *Nat. Mater.* **7**, 960-965 (2008)

As stated on page 3 2nd paragraph, “...long-range order becomes more fragile and vulnerable to impurities [Nie_2014, Imry_1975, Cardy_1996].” Therefore, importance of suppressing disorder to stabilize C-CDW at room temperature cannot be understated.

[Nie_2014]: Nie, L. et al., *PNAS* **111**, 7980-7985 (2014)
[Imry_1975]: Imry, Y. et al., *Phys. Rev. Lett* **35**, 1399-1401 (1975)
[Cardy_1996]: Cardy, J. “*Scaling and renormalization in statistical physics*” (1996)

The authors must consider the effect of CDW harmonics in their free energy analysis. Otherwise, whether the tC-CDW phase is more stable than the NC phase (or other phases) is ignored intentionally, which cannot be accepted. Therefore, why the NC phase was not observed must be clarified based on the previous understanding of 1T-TaS₂.

Robert Hovden, PhD
Assistant Professor
Materials Science & Eng.
University of Michigan
hovden@umich.edu
hovdenlab.com
c. 770-265-4042

Our manuscript shows experimentally that NC phase was not observed. Instead, a twinned commensurate CDW is present. As argued earlier for major comment 4, the role of the computational analysis is to capture the twinning which had not been discussed by previous Landau theory papers, rather than to make claims of stability of tC-CDW over the NC phase.

Note also that one cannot distinguish between C and NC by resistance measurement, and diffraction images must be considered.

We agree with the reviewer that C and NC can't be distinguished solely by resistance measurements. The diffraction data that we presented (Fig. 1d, 2a, S5c S12) clearly demonstrates that our polytypic heterostructure hosts C-CDW at room temperature. In addition, Fig. S10 shows mixed (NC + C) state formed by partial conversion of polytypes---clear demonstration that we can distinguish NC and C phase.

Minor Comments

1. In the introduction the authors write "1T polytype ... exhibits three distinct CDW phases: commensurate (C), nearly-commensurate (NC), and incommensurate (IC)" but this is wrong. There is also the triclinic (quasi-stripe) phase in 1T-TaS₂: See X-ray experiment [Synthetic Metals 11, 85 (1985)], STM experiment [Adv. Phys 37, 559 (1988)], and recent development such as [Sci. Rep. 10, 1239 (2020)], [Sci. Rep. 9, 7066 (2019)], and [npj Quantum Mater. 4, 32 (2019)]. There is also the stripe phase in 2H-TaSe₂. These phases cannot be ignored, and they must be discussed, including why they do not appear in Fig. 4 a.

We have noted the triclinic phase in 1T-TaS₂ on page 1, para 3 with the suggested reference so it now reads, "Octahedral (Oc) coordination found in the metastable 1T polytype has inversion symmetry and exhibits three distinct, salient CDW phases: commensurate (C), nearly-commensurate (NC), and incommensurate (IC). An intermediate triclinic phase has also been reported [Tanda_1985, Coleman_2006, Nakatsugawa_2020, Wang_2019, Gerasimenko_2019]. On page 4, para 3 "...tC \rightleftharpoons IC transition is reversible and the NC-CDW phase is removed. Note that the intermediate triclinic phase was also not observed..."

[Tanda_1985]: Tanda, S. and Sambongi, T. *Synth. Met.* **11**, 85-100 (1985)

[Coleman_2006]: Coleman, R. V. et al., *Adv. Phys.* **37**, 559-644 (2006)

[Nakatsugawa_2020]: Nakatsugawa, K. et al., *Sci. Rep.* **10**, 1239 (2020)

[Wang_2019]: Wang, W. et al., *Sci. Rep.* **9**, 7066 (2019)

[Gerasimenko_2019]: Gerasimenko, Y. A. et al., *npj Quantum Mater.* **4**, 32 (2019)

2. The C-CDW transition temperature increases drastically when the sample thickness decreases to a few layers. For instance, the C-CDW phase is obtained in tri-layer 1T-TaS₂ above room temperature (Ref. [38]). For this reason, the authors write "free-standing ultrathin 1T-TaS₂ degrades long-range order [38]" but this is wrong at least for three layers. Therefore, the authors should discuss the increase of transition temperature in comparison with these facts.

Robert Hovden, PhD
Assistant Professor
Materials Science & Eng.
University of Michigan
hovden@umich.edu
hovdenlab.com
c. 770-265-4042

Sakabe et al., [38] is a nice work that reveals CDW behavior in 1, 2, and 3 layers 1T-TaS₂. While it is true that C-CDW was obtained in 3-layer system, the authors state that the correlation length of the CDW is rather short (<10 nm) [38, Table 1]. Our polytype heterostructure maintained commensuration for at least (1 μm)² of area. Therefore, this is in agreement to our claim that encapsulation of ultrathin TaS₂ in native chemical and epitaxial environment increases transition temperature.

3. *We get a strong impression from Fig. 1 (a) that α/β appear alternately, which is highly unlikely to occur.*

We appreciate the reviewer's concern. Fig. 1a is a schematic diagram that represents one of possible configuration and there is 1/4 chance of having shown structure. We have now added Fig. S1 that shows a schematic with larger fields of view and randomized α/β selection. On page 1, para 4 we reference this newly created SI, Fig. S1 that shows a schematic with larger fields of view and randomized α/β selection. "...illustrated in Figure 1a (See also: Fig. S1). The caption for Fig S1 reads "Schematic Diagram of tC-CDW in Interleaved Polytypic Heterostructure a) Octahedral layers are sparsely interleaved in between metallic prismatic layers and hosts a C-CDW. Colored overlay represents α (blue) and β (magenta) C-CDW hosted in each octahedral layer. This schematic represents one possible configuration where α and β CDW orientation in each layer occurs with a random, equal likelihood."

4. *In Fig. 1 (d), the raw diffraction image of tC-CDW must be shown first without red and blue points (is it shown in FIG. S3?)*

We agree that showing the raw, non-annotated data is important. On page 12 of Supplemental Information, we have added Figure S12 which is the raw diffraction image.

5. *The authors write "Here, each 2D 1T-TaS₂ CDW is in its native chemical, epitaxial, and unstrained environment." Does "epitaxial" mean molecular beam epitaxy or something else?*

Here, epitaxial means that 1T-TaS₂ (i.e. octahedral layers) are encapsulated with well-defined orientation and registry to the prismatic layers.

6. *In the "Synthesis of 1T-TaS_xSe_{2-x} crystal" section in Methods, the temperature unit is not shown correctly.*

Page 5, "Synthesis of 1T-TaS_xSe_{2-x} crystal now reads" "... where the temperature of source zone and growth zone was fixed at 950 °C and 850 °C, respectively."

Therefore, we will reconsider whether the manuscript can be published after the above issues are clarified. The manuscript needs a major reconsideration based on these comments. Unfortunately, it must be said that the basic understanding of CDW in 1T-TaS₂ is lacking.

We thank the reviewer for their detailed feedback, which has improved our manuscript.

Robert Hovden, PhD
Assistant Professor
Materials Science & Eng.
University of Michigan
hovden@umich.edu
hovdenlab.com
c. 770-265-4042

We feel that these changes have improved the manuscript and thank the referees for the helpful suggestions. Please do not hesitate to contact us with any further comments or requests.

Sincerely,

Robert Hovden, Ph.D.

REVIEWER COMMENTS

Reviewer #1 (Remarks to the Author):

My opinion in the first round was that the paper showed some impressive results, but I thought the abstract and some of the claims were a bit over-optimistic. I gave some suggestions for how I thought the paper should be improved. They've addressed most of my concerns well. In particular, I like that they have changed the title and I think the new wording is more accurate.

The one thing I'm still a bit confused over is their use of "clean". The problem is that "clean" can refer to both the carrier mean free path and to the general cleanliness of sample preparation. Given the repeated use of "clean", I really think they mean something to do with the coherence length. Anyway, that's just a wording issue so if they can clarify that, they've answered all my points.

The only other question is about whether they've unambiguously identified the charge-density-wave phase. Honestly, I'm not sure, but I think they've made a convincing argument and this work will be a useful publication on this topic.

Reviewer #2 (Remarks to the Author):

I have carefully read the revised manuscript and the Authors' response to my comments as well as to the insightful and rigorous evaluations of the other Referees. I did not think that the manuscript had very much that needed to be improved upon, but nevertheless the Authors have made a number of informative and clarifying revisions. I especially appreciate the addition of Figs S3 and S12 and accompanying discussions. I am satisfied that the Authors have appropriately and convincingly responded to nearly all comments and criticisms.

I just have one second-order perturbation which must be seriously considered, concerning a comment from Referee 1, and the Authors' response.

Referee 1's comment:

“The title is reasonable, but for my taste the abstract is a little too overblown. I didn’t like the “only in unrealized clean limit 2D materials such as 1T-TaS2” – I’m not sure the material is “unrealized”, nor that it the only such candidate material. Also I think “engineering” is not an accurate description for heat treatment, unless they demonstrated, say, really precise control of the number and type of layers.”

To which the Authors reply (in part):

“... In [Yu_2015], Yu et al established that 1T-TaS2 does not support CDW structure in low-dimensions.

[Yu_2015]: Yu, Y. et al., Nat. Nanotechnol. 10, 270-276 (2015) ...”

Now I realize that the Authors were under the impression that in thin 1T-TaS2, the CCDW is not just somehow dirty or disordered, but actually does not exist.

I would like to point out that 1T-TaS2 definitely does support a CCDW even in the monolayer limit, as evidenced using STM. (See for example Vano et al., arXiv: 2103.11989, and note that the same is true for 1T-TaSe2, as seen in Chen et al., Nature Physics 16, 218–224 (2020). It is interesting that this reveals a potential contradiction in the literature.)

Consequently, I think the first sentence of the Abstract, particularly the word “unrealized” needs to be reconsidered, as well as the following sentence in the Introduction: “Unfortunately, extrinsic and thermal disorder in free standing 2D layers degrades correlation--driven quantum behavior [16, 17] and clean 2D charge density waves or superconductivity are near absent [18].”

I don’t think this development significantly diminishes the message of this manuscript or counts against its clear novelty and usefulness. Therefore, after this minor revision, I think the manuscript should be accepted.

Reviewer #3 (Remarks to the Author):

We read carefully over the authors’ replies and the citations they give to confirm their arguments. We highly appreciate the new experimental techniques and results. However, for the interpretation/theory part, many replies are not satisfactory because either i) comments are not answered, ii) incorrect, iii) replies are inconsistent with the citations, or even iv) self-contradictory.

Therefore, this manuscript cannot be published to Nature Communications in the present status: We must reject this manuscript unless a major reconsideration is made.

Major Comment 1:

We agree with the replies to this comment. We now understand the novelty of the authors' experimental method. The authors clearly demonstrate that they did not observe in-plane α/β multidomain and α/β are randomly obtained in each layer. However, this reply only partially answers our concern. We asked, "Layers with α -CCDW only or β -CCDW have not been observed so far." So, can the author estimate (both for academic and practical purpose) if they obtain a domain only or β domain only in each layer, or if they still get "large" α/β domains.

Major Comment 2:

There seems to be a miscommunication here, so we ask the same questions in a different form.

1. We understand the authors argue that their samples show an interleaved polytype, not the ordered 1T, 2H, or 4Hb polytypes. However, whether "4Hb layers" appear or not cannot be decided from the contrast in the TEM image. This is also what we meant by "4Hb may also appear in Fig2i". Diffraction pattern in 4Hb is similar to the authors' result of tC peaks, so this important possibility must be discussed.

2. We understand the discussion on streak in Fig.S8b. However, Fig.S8a and Fig.S8b still look similar, and the difference may be just a matter of intensity. For example, a chain of peaks is clearly visible along [0001] in Fig. S8a. Increasing the intensity of these peaks can lead to a streak-like pattern. The same chain structure appears also in low-intensity regions (e.g. left of Fig. S8b) which are basically identical to the [0001] direction in Fig. S8b. In addition, contrary to Fig. S8b, the streaks must appear also at the low-density regions. So, because important information in this kind of experiment is not intensity but peak position and their half-width, Fig.S8a and Fig.S8b seem to be equivalent. For this reason, the authors write "4Hb polytype exhibits superlattice peaks along [0001] that corresponds to a 4-layer repeat" but the difference from this chain of peaks must be clarified.

There is another problem with Fig. S8b. Usually, the height of NC-CDW peak ($c^*/3$) in bulk 1T-TaS₂ is due to three-layer stacking of 1T layers. However, In Fig. S8b, CDW peaks appear at the same height, which is problematic for the following reasons. 1) In contrary to Fig. 2i and S7, a large portion of 1T polytype with three-layer stacking may have remained in the heat-treated samples. How many 1T layers are in the sample used in Fig. S8b? 2) In contrary to the authors' argument, NC may have remained in the heat-treated samples.

In addition, Fig. 2i and S7 are hard to believe as evidence for interleaved polytype unless similar images without Se doping are provided (see Major Comment 3).

3. If 1T, 2H, and 4Hb "layers" are all present, then each layer can have an effect on the electric resistance. For example, the authors reply that

"4Hb-TaS₂ features a vertical resistance jump at 85°C and very sudden change of slope at 20 K. Our polytypic heterostructure shows a mild slope at ~85°C and gradual change of slope at ~50 K."

But these "slopes" may appear because 4Hb layers are present.

Major Comment 3:

The following questions were not answered:

- "the authors must explain clearly if they obtain a similar image [namely Fig. 2 (h,i)] without Se doping"

- "2H-TaSe₂ which has different CDW phases and different superlattice vectors can also appear"

We understand that the authors used Ta_xSe_{2-x} only for the convenience of HAADF-STEM micrographs. We would still ask the authors to provide the result of HAADF-STEM micrographs without Se doping because it is important evidence for the interleaved polytype: It is difficult to believe their arguments otherwise. If HAADF-STEM is impossible without Se doping, the authors must mention it clearly.

Major Comment 4:

The replies to this comment are problematic, so we highly recommend removing the free-energy section from the manuscript because this model is not even minimal. We are afraid that this section will decrease the rating of the authors' experimental results. If the authors still want to include this section, then the following comments, which are based on their reply, must be answered carefully.

- The authors write "A recent study ([Zong_2018]) used a similar argument..." but this is incorrect. [Zong_2018] uses the standard McMillan-Nakanishi-Shiba-type CDW free energy theory, as it can be verified by the supplementary material of [Zong_2018] and references therein. The position of CDW local minima is given by CDW reciprocal lattice vectors (or their norm in the present case), not by "CDW orientations". The CDW orientation is not enough as a CDW order parameter, because CDWs with the same angle but with different norms will all have the same energy. C and IC have different norms, so the authors' free energy cannot describe the IC phase. For example, Figure 3 b) is incorrect and it cannot explain the diffuse IC peak in Figure 2 a).

- For our comment "It is not clear what the hexagonal grid in Fig3 represents" the authors reply that "Each hexagonal cell represents a Ta ion site." Does "Ta ion site" mean the central Ta on a David star unit? Because the authors are discussing CDWs, the sites must be at least as large as one David star unit. Moreover, because different CDW phases have different norms, fixing the size of a hexagonal site is not appropriate.

- Then, the authors write "Here, we show that kinetic means (quench) alone can lead to the formation of twins." This statement is also incorrect. A comparison with the result of [Zong_2018] cannot be made unless the same kind of free energy is used.

- "instead of a full characterization that includes all these ingredients ... our objective here is to demonstrate a minimum phenomenological model which incorporates a few key ingredients." We are not asking to include all the "ingredients". A model analogous to [Zong_2018] can be such "minimum model". The free energy in the present manuscript is far from minimal.

- "This approximation correctly captures qualitative features away from the IC-CDW transition." This sentence is self-contradictory. With the authors' approximation, it is not appropriate to argue that the random- θ state corresponds to the IC phase since the IC phase cannot be described correctly in the first place and C/IC have different wave vectors. For example, Figure 3 b) is inappropriate and cannot explain the diffuse IC peak in Figure 2 a).

- Even if the authors can refute the comments given above, they do not include the following information required to understand their result.

1. In the main article: i) the values of a_2, a_4 , and T_c , ii) what temperature they give for “High T” and “Low T” in Figure 3, and iii) what (average) value of wave vector q they used to plot Figure 3.

2. In appendix: A_2, A_4, γ are not defined.

-Reply on diffuse peaks in the IC phase: Unfortunately, the reply does not answer our concern. We have specifically asked about the “IC phase in clean TaS₂ single crystals”. We also implicitly meant “in equilibrium” as is the case of the present manuscript. Indeed, [Vogelgesang_2017] and [Han_2015] consider CDW in optically excited TaS₂. However, the physics of non-equilibrium (transient) systems is different from the physics of equilibrium systems. For instance, in [Vogelgesang_2017], the IC peak after excitation has a finite width but this width vanishes at large time, in contrast to the authors’ claim. It might be possible to extend the interpretation in [ElBaggari_2018] to TaS₂, but this would require careful investigation.

-Moreover, the authors explain the diffuse IC peak with thermal disorder: This is also questionable because the authors show the IC Bragg peaks only for high temperature, $T=450^\circ\text{C}$ (or is $T=450^\circ\text{C}$ a mistake and the correct temperature is $T=450\text{K}$?). Are they still diffuse above the IC-tC transition temperature around $T=350\text{K}$?

Major Comment 5:

The NC phase is not a disordered phase. Disorder is not significant in large single crystals. For example, [S. Tanda and T. Sambongi, *Synthetic Metals*, 11 (1985) 85-100] report the correlation of NC phase along the c-axis for a thousand of layers. Although the connection between disorder and the stability of the NC phase should be investigated in optically excited systems, thin-film, or on the surface of a bulk crystal (where melting of CDW can occur), NC is still a legitimate phase at thermal equilibrium.

Therefore, the use of the word “disorder” needs caution. The authors write in the introduction “Metallic layers screen impurity potentials to suppress the disordered nearly-commensurate (NCCDW) phase” but such sentence is not acceptable unless they explain what “disordered” means in this manuscript. Sentences like “impurity and disorder decrease NC transition temperature and stabilize the C phase” are acceptable, but merely writing “disordered NC phase” is not acceptable.

In addition, in the “Clean Polytype Heterostructures” section, the authors write “When the disorder strength reaches a certain threshold, the long-range CCDW phase gives its way to a disordered phase [Nie_2014]. Here, each 2D 1T-TaS₂ CDW is in its native chemical, epitaxial, and unstrained environment” but there is a leap of logic because [Nie_2014] discusses cuprates. It is not an appropriate citation to argue that the NC is TaS₂ is disordered.

Minor Comment 4:

Epitaxy usually means molecular beam epitaxy (MBE). If the authors did not use MBE, then the word “epitaxy” cannot be used. Note the Greek origin of the word “epitaxy”: “epi” means “on top” and “taxy” means “in an ordered manner”

So, it is better to use other words like “intergrowth”, or as Wilson uses, intrapolytypic.

Additional comment: We highly recommend that the authors fix their temperature unit: use either degree Celsius or Kelvin.

Therefore, the manuscript needs a major reconsideration based on these comments.

Robert Hovden, PhD
Assistant Professor
Materials Science & Eng.
University of Michigan
hovden@umich.edu
hovdenlab.com
c. 770-265-4042

Response to reviewers

Reviewer #1:

My opinion in the first round was that the paper showed some impressive results, but I thought the abstract and some of the claims were a bit over-optimistic. I gave some suggestions for how I thought the paper should be improved. They've addressed most of my concerns well. In particular, I like that they have changed the title and I think the new wording is more accurate.

We appreciate the response and glad to hear our revisions are well received.

The one thing I'm still a bit confused over is their use of "clean". The problem is that "clean" can refer to both the carrier mean free path and to the general cleanliness of sample preparation. Given the repeated use of "clean", I really think they mean something to do with the coherence length. Anyway, that's just a wording issue so if they can clarify that, they've answered all my points.

To clarify, page 1, para 2 of Introduction, now reads "...by synthesizing clean (minimal disorder, impurity, or defects) ...".

The only other question is about whether they've unambiguously identified the charge-density-wave phase. Honestly, I'm not sure, but I think they've made a convincing argument and this work will be a useful publication on this topic.

Robert Hovden, PhD
Assistant Professor
Materials Science & Eng.
University of Michigan
hovden@umich.edu
hovdenlab.com
c. 770-265-4042

Reviewer #2:

I have carefully read the revised manuscript and the Authors' response to my comments as well as to the insightful and rigorous evaluations of the other Referees. I did not think that the manuscript had very much that needed to be improved upon, but nevertheless the Authors have made a number of informative and clarifying revisions. I especially appreciate the addition of Figs S3 and S12 and accompanying discussions. I am satisfied that the Authors have appropriately and convincingly responded to nearly all comments and criticisms.

We appreciate the reviewer's positive remarks; we agree that the manuscript is suitable for publication.

I just have one second-order perturbation which must be seriously considered, concerning a comment from Referee 1, and the Authors' response.

Referee 1's comment:

"The title is reasonable, but for my taste the abstract is a little too overblown. I didn't like the "only in unrealized clean limit 2D materials such as 1T-TaS₂" – I'm not sure the material is "unrealized", nor that it the only such candidate material. Also I think "engineering" is not an accurate description for heat treatment, unless they demonstrated, say, really precise control of the number and type of layers."

To which the Authors reply (in part):

"... In [Yu_2015], Yu et al established that 1T-TaS₂ does not support CDW structure in low-dimensions. [Yu_2015]: Yu, Y. et al., Nat. Nanotechnol. 10, 270-276 (2015) ..."

Now I realize that the Authors were under the impression that in thin 1T-TaS₂, the CCDW is not just somehow dirty or disordered, but actually does not exist.

I would like to point out that 1T-TaS₂ definitely does support a CCDW even in the monolayer limit, as evidenced using STM. (See for example Vano et al., arXiv: 2103.11989, and note that the same is true for 1T-TaSe₂, as seen in Chen et al., Nature Physics 16, 218–224 (2020). It is interesting that this reveals a potential contradiction in the literature.)

Consequently, I think the first sentence of the Abstract, particularly the word "unrealized" needs to be reconsidered, as well as the following sentence in the Introduction: "Unfortunately, extrinsic and thermal disorder in free standing 2D layers degrades correlation-driven quantum behavior [16, 17] and clean 2D charge density waves or superconductivity are near absent [18]."

To avoid conflict, we have removed "unrealized" from the abstract.

We thank the reviewer for bringing these references to our attention and highlighting the potential contradiction in the literature. We would like to note the CCDWs realized in monolayer TaS₂. Sakabe et al. [npj Quantum Mater. 2, 22 (2017)] had shown that monolayer TaS₂ exfoliated from a bulk crystal shows extremely short correlation length of ~7.5 Å. Vaňo et al.'s very recent work is impressive and shows MBE grown monolayers with larger CCDW domains c.a. (30nm)². We note, this

Robert Hovden, PhD
Assistant Professor
Materials Science & Eng.
University of Michigan
hovden@umich.edu
hovdenlab.com
c. 770-265-4042

manuscript has been in review at Nature Comm since Feb 17, 2021 and Vaño et al. submitted to Arxiv after our submission have not been considered.

I don't think this development significantly diminishes the message of this manuscript or counts against its clear novelty and usefulness. Therefore, after this minor revision, I think the manuscript should be accepted.

We again appreciate the positive response.

Robert Hovden, PhD
Assistant Professor
Materials Science & Eng.
University of Michigan
hovden@umich.edu
hovdenlab.com
c. 770-265-4042

Reviewer #3:

We read carefully over the authors' replies and the citations they give to confirm their arguments. We highly appreciate the new experimental techniques and results. However, for the interpretation/theory part, many replies are not satisfactory because either i) comments are not answered, ii) incorrect, iii) replies are inconsistent with the citations, or even iv) self-contradictory. Therefore, this manuscript cannot be published to Nature Communications in the present status: We must reject this manuscript unless a major reconsideration is made.

We appreciate the time and effort Reviewer #3 has committed to our work—the manuscript has undoubtedly improved through discussion. We have conducted additional in-situ STEM measurements in SI Fig S13. It is with respect and care that we have modified our manuscript or stated our disagreement.

Major Comment 1:

We agree with the replies to this comment. We now understand the novelty of the authors' experimental method. The authors clearly demonstrate that they did not observe in-plane α/β multidomain and α/β are randomly obtained in each layer. However, this reply only partially answers our concern. We asked, "Layers with α -CCDW only or β -CCDW have not been observed so far." So, can the author estimate (both for academic and practical purpose) if they obtain a domain only or β domain only in each layer, or if they still get "large" α/β domains.

In all experiments, no noticeable in-plane twinning was observed. 4D-STEM allowed us to measure domains with ~ 4.6 nm resolution and across >1 μm fields of view. We investigated an estimated upwards of 100 μm^2 total area across multiple specimens. This combination of resolution (in real and k-space) across large fields of view revealed a surprising lack of spatial variation which leads us to believe that in-plane twinning is unlikely or the domains are larger than the areas we measured.

Major Comment 2:

There seems to be a miscommunication here, so we ask the same questions in a different form.

1. We understand the authors argue that their samples show an interleaved polytype, not the ordered 1T, 2H, or 4Hb polytypes. However, whether "4Hb layers" appear or not cannot be decided from the contrast in the TEM image. This is also what we meant by "4Hb may also appear in Fig2i". Diffraction pattern in 4Hb is similar to the authors' result of tC peaks, so this important possibility must be discussed.

We have conducted an additional in-situ STEM experiment (Supplemental Figure S13) that shows atomic structure of TaS₂ before and after the mixed polytype transition. We discuss this further in Major Comment #3.

Robert Hovden, PhD
Assistant Professor
Materials Science & Eng.
University of Michigan
hovden@umich.edu
hovdenlab.com
c. 770-265-4042

It is incorrect to state that the lack of a “4Hb phase is determined from a single TEM image”. Here we use a combination of TEM diffraction, time-resolved in-situ TEM, and STEM to observe the interleaved polytype. The 4Hb phase is structurally distinct—the diffraction data in Fig. S8 is definitive (as well as STEM Fig. 2i, j and Fig. S7). Moreover, we note the real-space CDW structure of 4Hb remains unreported—our work will certainly bring new understanding to future investigation of CDWs in polytypes.

2. We understand the discussion on streak in Fig.S8b. However, Fig.S8a and Fig.S8b still look similar, and the difference may be just a matter of intensity. For example, a chain of peaks is clearly visible along [0001] in Fig. S8a. Increasing the intensity of these peaks can lead to a streak-like pattern. The same chain structure appears also in low-intensity regions (e.g. left of Fig. S8b) which are basically identical to the [0001] direction in Fig. S8b. In addition, contrary to Fig. S8b, the streaks must appear also at the low-density regions. So, because important information in this kind of experiment is not intensity but peak position and their half-width, Fig.S8a and Fig.S8b seem to be equivalent. For this reason, the authors write “4Hb polytype exhibits superlattice peaks along [0001] that corresponds to a 4-layer repeat” but the difference from this chain of peaks must be clarified.

There is another problem with Fig. S8b. Usually, the height of NC-CDW peak ($c^/3$) in bulk 1T-TaS₂ is due to three-layer stacking of 1T layers. However, In Fig. S8b, CDW peaks appear at the same height, which is problematic for the following reasons. 1) In contrary to Fig. 2i and S7, a large portion of 1T polytype with three-layer stacking may have remained in the heat-treated samples. How many 1T layers are in the sample used in Fig. S8b?*

We do not agree with the referee’s interpretation of our diffraction data. Fig. S8a and Fig. S8b do not “look similar” and the difference is not “a matter of intensity”. Uncorrelated stacking changes / disorder manifests as streaks in reciprocal space along the Bragg peaks. In our work, the uncorrelated stacking from polytype transformation is consistent with the diffraction, real-space measurements, and in-situ TEM. To further support our claim, line profiles along c-axis are added as Fig. S8c. The line profiles clearly show an order of magnitude increase in streaking as well as clear absence of 4Hb specific diffraction peaks.

Integrity and honesty are essential to our work, we do not mislead readers with un-representative datasets by saturating, clipping, or manipulating our data. The contrast is directly comparable and clearly distinct. Raw data is always available upon request. Moreover, our work is reproducible and we encourage the referee and other scientists to reproduce this work as we have done many times.

2) In contrary to the authors’ argument, NC may have remained in the heat-treated samples.

We agree with the reviewer that under certain heat-treatments NC can remain. As reviewer #1 highlighted in the previous reviews, our manuscript also shows how one can partially convert layers to created mixed NC and tC CDW or a pure tC phase. On page 11 of Supplemental Information, Fig. S10 is annotated to highlight NC and C superlattice peaks. The caption reads “a) Room temperature SAED, taken after partial polytypic transitions have occurred, shows both α -NC and β -C CDW peaks

Robert Hovden, PhD
Assistant Professor
Materials Science & Eng.
University of Michigan
hovden@umich.edu
hovdenlab.com
c. 770-265-4042

where many adjacent (i. e. coupled) CDW layers remain alongside isolated 2D CDW layers, resulting in the presence of both NC and C CDW respectively...”.

In addition, Fig. 2i and S7 are hard to believe as evidence for interleaved polytype unless similar images without Se doping are provided (see Major Comment 3).

We conducted an additional in situ STEM experiment (Supplemental Figure S13) that shows atomic structure of TaS₂ before and after the tC-CDW, mixed polytype transition occurs. The results are consistent with the mixed polytype structure. As expected, based on our experiments and previous literature reporting the bulk polytype phase transition, the Se doped specimen is nearly identical to the undoped specimen. The reported phase transition in TaS₂ and TaSe₂ are around 600 K and 550 K which we also confirmed (multiple times) herein. In both systems, observed by in-situ TEM, the kinetics of transformation are indistinguishable. We have confirmed syntheses both in-situ TEM, diffraction, and ex-situ with comparable changes in optical absorption.

3. If 1T, 2H, and 4Hb “layers” are all present, then each layer can have an effect on the electric resistance. For example, the authors reply that “4Hb-TaS2 features a vertical resistance jump at 85 °C and very sudden change of slope at 20 K. Our polytypic heterostructure shows a mild slope at ~85 °C and gradual change of slope at ~50 K.” But these “slopes” may appear because 4Hb layers are present.

We think the coordination in each layer is the first order contribution to the electrical properties of the material. Coexisting presence of Octahedral and Prismatic characteristics explains our electrical measurements, as well as the optical measurements. This interpretation is consistent with Di Salvo et al. who, when discussing bulk 4Hb-TaS₂, wrote “Apparently, the individual layers in the 4Hb compound retain the electrical characteristics of the phase which contains only layers of that coordination” [Di Salvo et al., J. Phys. Chem. Solids, 34, 1357-1362 (1973)]. The key takeaway is that the heat-treated sample is not a bulk 4Hb-TaS₂ sample.

Major Comment 3:

The following questions were not answered:

- “the authors must explain clearly if they obtain a similar image [namely Fig. 2 (h,i)] without Se doping”*
- “2H-TaSe2 which has different CDW phases and different superlattice vectors can also appear”*

We believe the referee mis-understands our work with regards the interleaving of polytypes when stating “2H-TaSe₂ which has different CDW phases and different superlattice vectors can also appear”. Either the referee assumes that CDW superlattice vectors alter the polytype phase transition or the referee believes our work is studying 2H-TaSe₂ CDWs. We are not studying TaSe₂ CDWs in this

Robert Hovden, PhD
Assistant Professor
Materials Science & Eng.
University of Michigan
hovden@umich.edu
hovdenlab.com
c. 770-265-4042

work and the polytype phase transition does not change with Se doping [Wilson_1975]. The reported phase transition in TaS₂ and TaSe₂ are around 600 K and 550 K which we also confirmed (multiple times) herein. In both systems, observed by in-situ TEM, the kinetics of transformation are indistinguishable. Both are synthesized in-situ or ex-situ with comparable changes in optical absorption (also reported). We are then left to speculate how an additional STEM image would offer any assurances to this referee, however, we have conducted an additional in situ STEM experiment (Supplemental Figure 13) that shows atomic structure of TaS₂ before (Fig. S13a) and after the tC-CDW (Fig. S13c), mixed polytype transition occurs in plan-view. In Fig. S13a, All Ta-sites are aligned (so called AA stacking) in projection as expected for 1T-TaS₂. In contrast, Fig. S13b shows that Ta sites are not fully aligned; the 4Hb polytype should have fully aligned Ta sites and indistinguishable with 1T in projection.

We understand that the authors used TaS_xSe_{2-x} only for the convenience of HAADF-STEM micrographs. We would still ask the authors to provide the result of HAADF-STEM micrographs without Se doping because it is important evidence for the interleaved polytype: It is difficult to believe their arguments otherwise. If HAADF-STEM is impossible without Se doping, the authors must mention it clearly.

We have now conducted an additional in-situ STEM experiment (Supplemental Figure S13) that shows atomic structure of TaS₂ before and after the tC-CDW, mixed polytype transition occurs in plan-view. The results are consistent with the mixed polytype structure and clearly shows that polytype structure we synthesized is distinct from 4Hb polytype. As expected, based on our experiments and previous literature reporting the bulk polytype phase transition, the Se doped specimen is nearly identical to the undoped specimen.

High angle annular dark field (HAADF) STEM provides atomic images through elastic scattering processes (Rutherford scattering) that create contrast proportional to the squared atomic number (number of protons²). In this way, HAADF is more sensitive to heavy elements. The new experiment and figure we now include (SFig. S13) is a plan view of TaS₂ where the Ta atoms dominate the image contrast. This plan view allows us to directly see the shift of metal (Ta) sites after thermal treatment.

Major Comment 4:

The replies to this comment are problematic, so we highly recommend removing the free-energy section from the manuscript because this model is not even minimal. We are afraid that this section will decrease the rating of the authors' experimental results. If the authors still want to include this section, then the following comments, which are based on their reply, must be answered carefully.

- The authors write "A recent study ([Zong_2018]) used a similar argument..." but this is incorrect. [Zong_2018] uses the standard McMillan-Nakanishi-Shiba-type CDW free energy theory, as it can be verified by the supplementary material of [Zong_2018] and references therein. The position of CDW local minima is given by CDW reciprocal lattice vectors (or their norm in the present case), not by "CDW orientations". The CDW orientation is not enough as a CDW order parameter, because CDWs

Robert Hovden, PhD
Assistant Professor
Materials Science & Eng.
University of Michigan
hovden@umich.edu
hovdenlab.com
c. 770-265-4042

with the same angle but with different norms will all have the same energy. C and IC have different norms, so the authors' free energy cannot describe the IC phase. For example, Figure 3 b) is incorrect and it cannot explain the diffuse IC peak in Figure 2 a).

Indeed, there is a body of theoretical investigations of the formation of the various charge density waves (IC, NC, C) in TaS₂. The free energies use a complex order parameter $A \cdot \exp[i\phi]$ where A is the amplitude and ϕ is the phase. However, there have been no studies (Landau models) that lead to the formation of twins. Typically, a single twin configuration is assumed [McMillan_1975, Nakanishi_1984, etc.], followed by expansions in terms of the amplitude and phase and coupling terms. A complete model for CDW twinning requires more robust theoretical work, an opinion we share with the referee. Our work only claims to provide a simple, minimal phenomenological model to qualitatively capture the formation of twin which we think is illustrative. Capturing the other details like the norm will require the additional terms used in the well-established theories in addition to the terms capturing the twinning, an undertaking well beyond the scope of these experimental reports.

We now state in the main manuscript (page 3, para 4) "This provides a simple, minimal phenomenological model to qualitatively capture the formation of twinned CDWs but does not model all remaining components of the complete CDW order parameter (See Methods)" as well as a more detailed discussion in the methods (added in the previous referee response). We also now state the model "qualitatively reproduces diffraction patterns".

- For our comment "It is not clear what the hexagonal grid in Fig3 represents" the authors reply that "Each hexagonal cell represents a Ta ion site." Does "Ta ion site" mean the central Ta on a David star unit? Because the authors are discussing CDWs, the sites must be at least as large as one David star unit. Moreover, because different CDW phases have different norms, fixing the size of a hexagonal site is not appropriate.

We respectfully disagree that when "discussing CDWs, the sites must be at least as large as one David star unit". We believe a CDW order parameter can have sub-supercell fluctuations. In either case, the norms are not essential to this illustrative model nor the main results of this manuscript.

- Then, the authors write "Here, we show that kinetic means (quench) alone can lead to the formation of twins." This statement is also incorrect. A comparison with the result of [Zong_2018] cannot be made unless the same kind of free energy is used.

This statement does not appear in our manuscript nor does a direct comparison with [Zong_2018].

- "instead of a full characterization that includes all these ingredients ... our objective here is to demonstrate a minimum phenomenological model which incorporates a few key ingredients." We are not asking to include all the "ingredients". A model analogous to [Zong_2018] can be such "minimum model". The free energy in the present manuscript is far from minimal.

Robert Hovden, PhD
Assistant Professor
Materials Science & Eng.
University of Michigan
hovden@umich.edu
hovdenlab.com
c. 770-265-4042

The reviewer states that “[Zong_2018] uses the standard McMillan-Nakanish-Shiba-type CDW free energy theory, as it can be verified by the supplementary materials”. However, the order parameter Zong et al. used represents the twin and not the full complex-valued, multi- q CDW order parameter as they describe in their Supplementary Materials. Notation-wise they used ψ which may have caused confusion. Zong et al. continues with “For simplicity, we limit ourselves to the discussion of breaking mirror symmetry only; a complete treatment of the phase transition involving a triple- q charge density modulation and discommensuration has been well formulated elsewhere.”

Ours goes one step further by explicitly describing the wavevector associated with each twin. Our manuscript states (page 6), “In general, a CDW is characterized by two intertwined ingredients: (1) the amplitude/phase of the CDW order parameter and (2) the length/orientation of the CDW wavevector. The significant role of the first ingredient has been extensively studied in literature [21, 32, 53, 54]. For this study, the second ingredient is the key focus, which characterizes the fluctuation in the wavevector orientation θ as a minimum model to characterize the competition between the two phases observed in the experiments. This approximation correctly captures qualitative features away from the IC \rightarrow C CDW transition. Near the phase transition, where additional fluctuations are no longer negligible, a more sophisticated model becomes needed.”

- *“This approximation correctly captures qualitative features away from the IC-CDW transition.” This sentence is self-contradictory. With the authors’ approximation, it is not appropriate to argue that the random- θ state corresponds to the IC phase since the IC phase cannot be described correctly in the first place and C/IC have different wave vectors. For example, Figure 3 b) is inappropriate and cannot explain the diffuse IC peak in Figure 2 a).*

We do not describe the IC phase as a random- θ but rather that there are fluctuations about a mean value. From structural diffraction analysis alone (no knowledge of the underlying energy landscape), the spatially diffuse IC peaks along the azimuthal directions in Figure 2a can be interpreted as (i. e. is consistent with) orientational variation in the CDW wavevector.

- *Even if the authors can refute the comments given above, they do not include the following information required to understand their result.*

1. *In the main article: i) the values of a_2, a_4 , and T_c , ii) what temperature they give for “High T ” and “Low T ” in Figure 3, and iii) what (average) value of wave vector q they used to plot Figure 3.*

The simulation parameters were already reported in SI9. High T means sufficiently above T_c .
iii) Our model is agnostic to the length of q -vector length.

2. *In appendix: A_2, A_4, γ are not defined.*

$A_2 = 1, A_4 = 10, \gamma = 0.5, 10^{10}$ iterations. The values were already reported in SI 9, however we now reference Supplemental Figure S9 in the appendix.

Robert Hovden, PhD
Assistant Professor
Materials Science & Eng.
University of Michigan
hovden@umich.edu
hovdenlab.com
c. 770-265-4042

The 2nd paragraph of “Modelling C \leftrightarrow IC transition now reads “... (See Supplemental Figure S9 for simulation parameters)”

-Reply on diffuse peaks in the IC phase: Unfortunately, the reply does not answer our concern. We have specifically asked about the “IC phase in clean TaS₂ single crystals”. We also implicitly meant “in equilibrium” as is the case of the present manuscript. Indeed, [Vogelgesang_2017] and [Han_2015] consider CDW in optically excited TaS₂. However, the physics of non-equilibrium (transient) systems is different from the physics of equilibrium systems. For instance, in [Vogelgesang_2017], the IC peak after excitation has a finite width but this width vanishes at large time, in contrast to the authors’ claim. It might be possible to extend the interpretation in [ElBaggari_2018] to TaS₂, but this would require careful investigation.

The IC peaks we observe in 1T-TaS₂ match that of all previous reports. The bulk samples are “clean” by common use of the word and the crystals are ordered. In this sense these IC peaks can be considered representative of bulk 1T-TaS₂. All diffraction measurements were sustainable over long periods of time (multiple days) and we do not consider transient behavior nor optical excitations.

-Moreover, the authors explain the diffuse IC peak with thermal disorder: This is also questionable because the authors show the IC Bragg peaks only for high temperature, $T=450\text{ }^{\circ}\text{C}$ (or is $T=450\text{ }^{\circ}\text{C}$ a mistake and the correct temperature is $T=450\text{K}$?). Are they still diffuse above the IC-tC transition temperature around $T=350\text{K}$?

IC ‘superlattice peak’ remain diffuse (especially compared to tC peaks) above 350K—becoming slightly more diffuse at a higher temperature.

Robert Hovden, PhD
Assistant Professor
Materials Science & Eng.
University of Michigan
hovden@umich.edu
hovdenlab.com
c. 770-265-4042

Major Comment 5:

The NC phase is not a disordered phase. Disorder is not significant in large single crystals. For example, [S. Tanda and T. Sambongi, Synthetic Metals, 11 (1985) 85-100] report the correlation of NC phase along the c-axis for a thousand of layers. Although the connection between disorder and the stability of the NC phase should be investigated in optically excited systems, thin-film, or on the surface of a bulk crystal (where melting of CDW can occur), NC is still a legitimate phase at thermal equilibrium.

Therefore, the use of the word “disorder” needs caution. The authors write in the introduction “Metallic layers screen impurity potentials to suppress the disordered nearly-commensurate (NCCDW) phase” but such sentence is not acceptable unless they explain what “disordered” means in this manuscript. Sentences like “impurity and disorder decrease NC transition temperature and stabilize the C phase” are acceptable, but merely writing “disordered NC phase” is not acceptable. In addition, in the “Clean Polytype Heterostructures” section, the authors write “When the disorder strength reaches a certain threshold, the long-range CCDW phase gives its way to a disordered phase [Nie_2014]. Here, each 2D 1T-TaS₂ CDW is in its native chemical, epitaxial, and unstrained environment” but there is a leap of logic because [Nie_2014] discusses cuprates. It is not an appropriate citation to argue that the NC is TaS₂ is disordered.

We have removed “disordered” on page 1, para 2 so that it reads, “Metallic layers screen impurity potentials to suppress the nearly-commensurate (NCCDW) phase”. Disorder is described later on page 1, “the conductive NC-CDW is generally accepted as a C-CDW with short range order [23–26] that permits electron transport along regions of discommensuration [27,28]”

We respectfully disagree with the referee’s statement, “the NC phase is not a disordered phase”. Growing views on charge order consider real-space fluctuations in the order parameter [A. Del Maestro et al., *Phys. Rev. B*, **74**, 024520 (2006), L. Nie et al., *PNAS*, **111**, 7980–7985 (2014)], I. El Baggari et al., *PNAS*, **115**, 1445–1450 (2018), Y. A. Gerasimenko et al., *npj Quantum Mater.* **4**, 32 (2019). Especially, STM image of NC-CDW in Fig. 2c of Gerasimenko et al. shows how disordered NC phase in TaS₂ is. We have added Gerasimenko et al. as reference in paragraph 3 of Introduction. The paragraph 3 now reads “... At room temperature, the conductive NC-CDW is generally accepted as a C-CDW with short range order [Gerasimenko et al., ...] ...” In reciprocal space these fluctuations can manifest as changes in the wave vector, cause peak intensity changes in first or second order peaks, and drive CDW phase transitions. The precise real-space structure of the NC-phase remains an active area of research.

Minor Comment 4:

*Epitaxy usually means molecular beam epitaxy (MBE). If the authors did not use MBE, then the word “epitaxy” cannot be used. Note the Greek origin of the word “epitaxy”: “epi” means “on top” and “taxy” means “in an ordered manner”
So, it is better to use other words like “intergrowth”, or as Wilson uses, intrapolytypic.*

Robert Hovden, PhD
Assistant Professor
Materials Science & Eng.
University of Michigan
hovden@umich.edu
hovdenlab.com
c. 770-265-4042

We have changed epitaxial to “endotaxial” which has some precedence in the literature [(Pancier et al. *Nat. Mater.* (2015), Biswas et al. *Nat. Chem.* (2011)].

Abstract: “...2D quantum states through endotaxial...”

Abstract: “This work introduces endotaxial polytype...”

Page 3, Para 3 “...in its native chemical endotaxial, and unstrained...”

Discussion: “...is in a chemically and endotaxially...”

Molecular beam epitaxy is a vacuum deposition method in which well-defined thermal beams of atoms or molecules react at a crystalline surface [1] which can produce an epitaxial film with proper conditions. Thus MBE is a deposition method, one like many others (sputtering, PLD, CVD) [2–4] that can achieve epitaxial film growth. Note that all of the cited works in the prior sentence use the word epitaxy and establish an epitaxial relationship between the film and substrate without using MBE as the deposition method. Colloquially, epitaxy refers to ordered atomic registry between materials and thus enhances the readability of our manuscript to the broad audience of Nature Communications.

[1]: APL Materials 3, 062403 (2015)

[2]: Science 249, 1549-1552 (1990)

[3]: Science 299, 1719-1722 (2003)

[4]: Nature 570, 91–95 (2019)

Additional comment: We highly recommend that the authors fix their temperature unit: use either degree Celsius or Kelvin.

All instances of Celsius are converted to Kelvin.

Fig. 2 Caption: “...~350 K ... ~620 K”

Appendix, Synthesis of 1T-TaS₂ Crystal: “...1170 K ...1220 K and 1120 K...”

Figure 2, S5, S7, S8, and S10 are modified to use Kelvin as well.

Therefore, the manuscript needs a major reconsideration based on these comments.

We sincerely thank Reviewer #3 for the time and rigor put into our manuscript. It has improved the presentation of our results.

Robert Hovden, PhD
Assistant Professor
Materials Science & Eng.
University of Michigan
hovden@umich.edu
hovdenlab.com
c. 770-265-4042

We feel that these changes have improved the manuscript and thank the referees for the helpful suggestions. Please do not hesitate to contact us with any further comments or requests.

Sincerely,

Robert Hovden, Ph.D.

REVIEWER COMMENTS

Reviewer #3 (Remarks to the Author):

In the previous comment we wrote that the manuscript is rejected unless a major reconsideration is made. Such reconsideration has not been made. Therefore, we must reject this manuscript. The main reasons to reject this article are as follows.

1. The authors do not want to reconsider their free-energy theory (Major Comment 4). Honestly, we do not understand why the authors want to keep their incorrect free-energy theory. If such oversimplified theory is published in a high-profile journal like Nature Communications, other researcher may use it and obtain false conclusions. There is a large experimental and theoretical background about why the McMillan-Nakanishi-Shiba theory (of course, including contribution from other researchers) is accepted as the standard free-energy theory of CDW phases in transition-metal dichalcogenides. Introducing an unjustified theory only for the convenience of the authors is not acceptable.

2. This reason is about Major Comments 2-3. Although we appreciate the additional experimental data, many questions are still not addressed properly. First, there seems to be several misunderstandings of our comments. Second, there are many questionable replies that we cannot agree. Third, we feel that some comments are not answered intentionally.

3. The authors give references to refute our comments (especially Major Comment 5), but these references are not appropriate: These references consider different materials or optically-excited/transient systems. It is not clear why these references can apply to TaS₂ at equilibrium. Moreover, this problem of citing inappropriate references appears repeatedly also in the manuscript. We commented on this issue but it was not addressed. Honestly, we feel that there is a problem of scientific inconsistency

In what follows we give comments which lead to our decision. We only consider the most problematic replies (except Major Comment 1).

Major Comment 1: We appreciate the replies for this comment. Once again, we highly appreciate the new experimental techniques and results.

Major Comment 2:

1. The main concerns in this comment were i) did the authors obtained 4Hb bulk, and ii) if they did not obtain 4Hb bulk, then can 4Hb layers appear in their samples.

For the first part, the authors have convinced that they did not obtain 4Hb bulk. Then, a natural question is can we still have some 4Hb layers in coexistence with other type of layers (second part). We do not understand why the authors ignore the possibility that 4Hb layers can appear.

2. If 1T, 2H, and 4Hb layers are in coexistence, then we can observe physical effects due to the 4Hb layers. In Fig S8c), the authors argue that they did not obtain 4Hb bulk. However, we can still observe signals which are possibly due to 4Hb layers (see supplementary figure where such possible peaks are highlighted by purple arrows).

3. We explicitly ask about possible contribution of 4Hb layers to the electric transport, but the authors conclude “The key takeaway is that the heat-treated sample is not a bulk 4Hb-TaS₂ sample” which is not the point of this comment.

Therefore, the authors consider only two possibilities. Either 4Hb is completely absent or the whole sample is all 4Hb (4Hb bulk). We do not understand why the authors are convinced that 4Hb layers cannot appear.

Additionally, we now understand that the NC phase can appear in the heat-treated sample.

Major Comment 3:

i) We now understand that HAADF-STEM micrograph images like Fig. 2h,i cannot be obtained without Se doping. The new experimental data supports the interleaved structure, but what kind of layers (1T, 2H, or 4Hb) can appear is still inconclusive. Moreover, as we asked in the previous round (not answered), how many 1T layers remain in the heat-treated samples is questionable. (There is also a problem with position of satellite peaks as commented in the previous round.)

ii) By “2H-TaSe₂ which has different CDW phases and different superlattice vectors can also appear” we meant whether or not 2H-TaSe₂ layers can appear in the TaS₁Se₁ sample. This possibility cannot be ignored because the same amount of S and Se are included. This may have consequence on the interleaved structure of TaS₁Se₁, hence the structure of heat-treated TaS₂ is still inconclusive from Fig. 2h,i.

Therefore, from Major Comment 2 and Major Comment 3, although the authors show that they did not obtain 4Hb bulk, we do not understand why the authors are convinced that only isolated layers

of 1T polytype remain. They ignore other possibilities, such as the presence of 4Hb layers which is suggested from Fig. S8 and from electric transport. This ignorance can lead to false conclusions.

Major Comment 4:

Once again, there are many reasons to consider the McMillan-Nakanishi-Shiba theory when discussing CDW in transition metal dichalcogenides (MX₂). We understand that some of the authors have experience of CDW or charge-order in square lattice. But, MX₂ has trigonal symmetry. Like Anderson discusses [Anderson, P. W., Basic Notions of Condensed Matter Physics, 40 (Addison-Wesley, 1984)], the triple-Q term is essential and cannot be ignored for the description of CDW phases in MX₂. This is a common sense in 2D CDW physics. In fact, by ignoring the triple-Q term one cannot explain the known CDW phases, including the NC phase and C phase.

Based on the manuscript and replies to our comment, we have to conclude that the authors do not understand CDW free energy, and there is no reason to accept their alternative theory.

Regarding Ref. [Zong_2018], the authors argue that a similar theory is used, but this is a misunderstanding. The order-parameter in ref. [Zong_2018] can be written as $\psi = \psi_\beta - \psi_\alpha$, where $\psi_{(\alpha/\beta)}$ are the complex scalar order parameters for the α/β -CCDW state. This may be an acceptable order parameter which breaks mirror symmetry. Nevertheless, citing Ref. [Zong_2018] is not a valid excuse to ignore triple-Q terms. If the authors are not capable to obtain a suitable free-energy, then they could have moved it to the appendix, like in Ref. [Zong_2018].

For the comment regarding the statement “This approximation correctly captures qualitative features away from the IC-CDW transition” we have missed the word “away from”. Also, by “random” we meant “fluctuating”. Our apologies for this confusion. However, the problem with diffuse IC-CDW peaks still remains.

For example, the authors write “The IC peaks we observe in 1T-TaS₂ match that of all previous reports” but we cannot agree on this statement. The IC peaks obtained by the authors are indeed diffuse. Diffuse peaks are really extended, which is typically due to structural imperfections. But if the satellite peak size is comparable to the primitive peaks from TaS₂ lattice, then the peaks are spot-like and not diffuse. Spot-like peaks mean long-range order of more than 100 layers or so. We can find spot-like IC-CDW peaks in 1T-TaS₂, such as Figure 2 in [G. Storeck et al., Structural Dynamics 7, 034304 (2020)] and Fig. 2b) and Fig. 3 c) in [Vogelgesang_2017] which was cited by the authors. Therefore, we find that the authors’ discussion on the IC phase is inconsistent with previous reports.

For these reasons, it is difficult to believe that the authors’ free energy theory qualitatively reproduces diffraction patterns.

Major Comment 5:

The authors do not understand the NC phase in 1T-TaS₂. They argue that the NC phase is a disordered phase. However, this argument is wrong for the system considered in this manuscript (namely TaS₂ at equilibrium). The authors give references to refute this comment, but these references are not appropriate: They consider different materials (square lattice) or optically-excited/transient systems. In fact, this problem of citing inappropriate references appears repeatedly in the manuscript. We commented on this issue but it was not addressed. (This problem appears again in the new reply. The authors clearly write "we do not consider transient behavior nor optical excitations" but they try to support their argument by citing work on optically-excited/transient systems).

We are aware of the work by Gerasimenko et al. about the so-called Hidden (H) CDW state which appear at highly non-equilibrium condition. They report the appearance of chiral order in a seemingly disordered H phase. In this reference, Fig. 2c (NC phase) is cited as "At elevated temperatures, $T > T_{C-NC} \sim 220$ K, the uniform C order changes to a modulated nearly commensurate (NC) CDW with a highly regular array of domains". Here, "highly regular array of domains" is used in contrast to the disordered domain-walls of the H-CDW phase. Please understand the references before citing them: It is not the work of the referees to do so.

Therefore, the authors write "At room temperature, the conductive NC-CDW is generally accepted as a C-CDW with short range order [Gerasimenko et al.,] ..." but this is incorrect. The NC-CDW phase and the H-CDW phase are different CDW phases.

Conclusion: We feel that there is a serious problem of scientific consistency in this manuscript. Also, many comments were not addressed properly. Therefore, we must reject this manuscript.

Robert Hovden, PhD
Assistant Professor
Materials Science & Eng.
University of Michigan
hovden@umich.edu
hovdenlab.com
c. 770-265-4042

Sept. 15, 2021

Response to Reviewers

We appreciate the feedback and positive support of Reviewers #1 and #2. Their input has improved the quality of our work.

Comments from Reviewer #3:

In the previous comment we wrote that the manuscript is rejected unless a major reconsideration is made. Such reconsideration has not been made. Therefore, we must reject this manuscript. The main reasons to reject this article are as follows.

1. The authors do not want to reconsider their free-energy theory (Major Comment 4). Honestly, we do not understand why the authors want to keep their incorrect free-energy theory. If such oversimplified theory is published in a high-profile journal like Nature Communications, other researcher may use it and obtain false conclusions. There is a large experimental and theoretical background about why the McMillan-Nakanishi-Shiba theory (of course, including contribution from other researchers) is accepted as the standard free-energy theory of CDW phases in transition-metal dichalcogenides. Introducing an unjustified theory only for the convenience of the authors is not acceptable.
2. This reason is about Major Comments 2-3. Although we appreciate the additional experimental data, many questions are still not addressed properly. First, there seems to be several misunderstandings of our comments. Second, there are many questionable replies that we cannot agree. Third, we feel that some comments are not answered intentionally.
3. The authors give references to refute our comments (especially Major Comment 5), but these references are not appropriate: These references consider different materials or optically-excited/transient systems. It is not clear why these references can apply to TaS₂ at equilibrium. Moreover, this problem of citing inappropriate references appears repeatedly also in the manuscript. We commented on this issue but it was not addressed. Honestly, we feel that there is a problem of scientific inconsistency.

In what follows we give comments which lead to our decision. We only consider the most problematic replies (except Major Comment 1).

Please see our point-by-point responses below. We respect every comment of the Reviewer.

Robert Hovden, PhD
Assistant Professor
Materials Science & Eng.
University of Michigan
hovden@umich.edu
hovdenlab.com
c. 770-265-4042

Major Comment 1: We appreciate the replies for this comment. Once again, we highly appreciate the new experimental techniques and results.

We appreciate the Reviewer's recognition of the additional experimental techniques and results produced during the review process.

Major Comment 2:

1. The main concerns in this comment were i) did the authors obtained 4Hb bulk, and ii) if they did not obtain 4Hb bulk, then can 4Hb layers appear in their samples. For the first part, the authors have convinced that they did not obtain 4Hb bulk. Then, a natural question is can we still have some 4Hb layers in coexistence with other type of layers (second part). We do not understand why the authors ignore the possibility that 4Hb layers can appear.

We're pleased the Reviewer is convinced the 4Hb phase of TaS₂ is not present to a significant degree in our heat-treated samples.

We appreciate the Reviewer's inquiry regarding local stacking sequences that may match the 4Hb phase. The clearest explanation of our system is to describe it as an uncorrelated and sparse interleaving of octahedral layers within many prismatic layers. As the layer-by-layer polytype transitions are uncorrelated, the final system is not well described by bulk polytype symmetry groups—including the periodic 4-layer unit cell called 4Hb. We agree that uncorrelated polytype stacking may permit a small (or even negligible) number of 'accidental' layers which locally match a four-layer 4Hb unit cell. Our work suggests that CDW states within octahedral layers are primarily influenced by the local environment that includes the polytype layers above and below—rather than long range correlated stacking order (or even correlation amongst 4 layers). We have created a finite set of polytype heterojunctions (Pr-Oc) that may share properties found in the 4Hb phase. Knowing what happens at the interface of polytype layers (as within the 4Hb phase) is a new direction of research; for which our manuscript becomes quite relevant.

We now make the Reviewer's comment apparent to readers on page 3, para. 1 by stating, "Although this system is best described as a sparse interleaving of Oc-layers within many Pr-layers (Fig. 2i), the uncorrelated polytype stacking may permit by chance a small (or even negligible) amount of layers which locally match a 4Hb (or another bulk polytype) unit cell."

2. If 1T, 2H, and 4Hb layers are in coexistence, then we can observe physical effects due to the 4Hb layers. In Fig S8c), the authors argue that they did not obtain 4Hb bulk. However, we can still observe signals which are possibly due to 4Hb layers (see supplementary figure where such possible peaks are highlighted by purple arrows).

Our experimental observations (Cross-sectional HAADF (Fig. 2i, S7b,c) and SAED (Fig. S8b,c)) definitively show that 4Hb (or other bulk polytypes) does not exist in any significant degree. The

Robert Hovden, PhD
Assistant Professor
Materials Science & Eng.
University of Michigan
hovden@umich.edu
hovdenlab.com
c. 770-265-4042

diffraction signals highlighted by the purple arrows tells the same story. The signals at the purple arrows (and gray dotted lines) are at or below the noise level and do not appear everywhere they are expected (e.g., between (0002) & (0003)). If 4Hb exists to any significant degree, the signal will have well-defined peaks at well-defined locations. On page 3, para 1 we now state "... the uncorrelated polytype stacking may permit by chance a small (or even negligible) amount of layers which locally match a 4Hb (or another bulk polytype) unit cell."

3. We explicitly ask about possible contribution of 4Hb layers to the electric transport, but the authors conclude "The key takeaway is that the heat-treated sample is not a bulk 4Hb-TaS₂ sample" which is not the point of this comment.

As addressed above, 4Hb layers does not exist in any significant degree in our heat-treated polytype heterostructure. Therefore, we should not attribute any transport effect to 4Hb layers.

Therefore, the authors consider only two possibilities. Either 4Hb is completely absent or the whole sample is all 4Hb (4Hb bulk). We do not understand why the authors are convinced that 4Hb layers cannot appear.

Our experimental investigations definitively show that 4Hb does not exist in any significant degree. However, we appreciate the Reviewer's concern. We now make the Reviewer's comment apparent to readers on page 3, para. 1 by stating, "Although this system is best described as a sparse interleaving of Oc-layers within many Pr-layers (Fig. 2i), the uncorrelated polytype stacking may permit by chance a small (or even negligible) amount of layers which locally match a 4Hb (or another bulk polytype) unit cell."

Additionally, we now understand that the NC phase can appear in the heat-treated sample.

Major Comment 3:

i) We now understand that HAADF-STEM micrograph images like Fig. 2h,i cannot be obtained without Se doping. The new experimental data supports the interleaved structure, but what kind of layers (1T, 2H, or 4Hb) can appear is still inconclusive.

Moreover, as we asked in the previous round (not answered), how many 1T layers remain in the heat-treated samples is questionable.

Describing any specific set of layers using bulk polytype nomenclature (1T, 2H, 4Hb...) confuses the main point of this manuscript. Our manuscript's cross-sectional HAADF-STEM (Fig. 2i, S7b,c) shows a representative stacking sequence. The thermally treated system consists of mostly prismatic layers encapsulating occasional octahedral layers—interleaved, sparse, mixed coordination.

Robert Hovden, PhD
Assistant Professor
Materials Science & Eng.
University of Michigan
hovden@umich.edu
hovdenlab.com
c. 770-265-4042

Because of the nature of layer-by-layer transition, there is no set number of “1T layers” for the heat-treated sample. As previously discussed, *the number of prismatic layers can be tuned*—and if many octahedral layers are retained, NC-CDW peaks are visible in SAED. On the other extreme, a long thermal treatment will convert all octahedral layers to prismatic layers. This manuscript focuses on sparse interleaving of octahedral layers within prismatic layers.

(There is also a problem with position of satellite peaks as commented in the previous round.)

To the best of our knowledge there was no mention of satellite peak position in the previous round.

ii) By “2H-TaSe₂ which has different CDW phases and different superlattice vectors can also appear” we meant whether or not 2H-TaSe₂ layers can appear in the TaS₁Se₁ sample. This possibility cannot be ignored because the same amount of S and Se are included. This may have consequence on the interleaved structure of TaS₁Se₁, hence the structure of heat-treated TaS₂ is still inconclusive from Fig. 2h,i.

Cross-sectional HAADF-STEM shows that there is no 2H-TaSe₂ layer in the thermally treated TaS₂ sample. In HAADF, TaSe₂ layers would appear noticeably brighter in comparison to TaS₂ layers—which we clearly do not observe. Our samples are composed of TaS₂ layers as reported.

Therefore, from Major Comment 2 and Major Comment 3, although the authors show that they did not obtain 4Hb bulk, we do not understand why the authors are convinced that only isolated layers of 1T polytype remain. They ignore other possibilities, such as the presence of 4Hb layers which is suggested from Fig. S8 and from electric transport. This ignorance can lead to false conclusions.

We appreciate the Reviewer’s inquiry regarding local stacking sequences that may match the 4Hb phase. We now make the comment apparent to readers on page 3, para. 1 by stating, “Although this system is best described as a sparse interleaving of Oc-layers within many Pr-layers (Fig. 2i), the uncorrelated polytype stacking may permit by chance a small (or even negligible) amount of layers which locally match a 4Hb (or another bulk polytype) unit cell.”

Major Comment 4:

Once again, there are many reasons to consider the McMillan-Nakanishi-Shiba theory when discussing CDW in transition metal dichalcogenides (MX₂). We understand that some of the authors have experience of CDW or charge-order in square lattice. But, MX₂ has trigonal symmetry. Like Anderson discusses [Anderson, P. W., Basic Notions of Condensed Matter Physics, 40 (Addison-Wesley,1984)], the triple-Q term is essential and cannot be ignored for the description of CDW phases in MX₂. This is a common sense in 2D CDW physics. In fact, by ignoring the triple-Q term one cannot explain the known CDW phases, including the NC phase and C phase.

Based on the manuscript and replies to our comment, we have to conclude that the authors do not understand CDW free energy, and there is no reason to accept their alternative theory.

Robert Hovden, PhD
Assistant Professor
Materials Science & Eng.
University of Michigan
hovden@umich.edu
hovdenlab.com
c. 770-265-4042

Regarding Ref. [Zong_2018], the authors argue that a similar theory is used, but this is a misunderstanding. The order-parameter in ref. [Zong_2018] can be written as $\psi = \psi_\beta - \psi_\alpha$, where $\psi_{(\alpha/\beta)}$ are the complex scalar order parameters for the α/β -CCDW state. This may be an acceptable order parameter which break mirror symmetry. Nevertheless, citing Ref. [Zong_2018] is not a valid excuse to ignore triple-Q terms. If the authors are not capable to obtain a suitable free-energy, then they could have moved it to the appendix, like in Ref. [Zong_2018].

We fully agree with the Reviewer that in systems with three/six fold rotational symmetry, the cubic term is the leading order non-linear term near the phase transition, and thus plays a very important there, prohibiting any second-order phase transitions. On the other hand, as we pointed out in the manuscript, our study and theory analysis only focus on the phase space deep inside the ordered phase, far away from the transition, where the cubic term is no longer the dominant term. Our intent is not to confuse or mislead readers. To make this more apparent, we have now moved the paragraph discussing the limitations to the first paragraph in the relevant methods section titled “Modelling C \Leftrightarrow IC transition” (page 6, para 4). We have removed the word “Landau,” so it reads “phenomenological model” on page 3, para 4.

We appreciate the reviewer’s expertise and desire for a comprehensive theory that captures CDW twinning in our system. Existing theories which include detailed properties of the CDW (such as the norm of the wavevector) are constrained by empirical observation and do not predict the twinning behavior we observe. Here two orientations of the CDW are observed in equal probability and we model the energy as a double well potential to qualitatively illustrate a reversible pathway between the twinned commensurate and IC states that is consistent with our observations. Our phenomenological approach is illustrative and shows qualitative agreement with diffraction measurements. For future studies, to better understand the phase transition and the regime close to the phase boundary, such a more comprehensive analysis is absolutely necessary, and we hope our study could help motivate efforts along that direction.

We apologize for causing too much focus on the work of Zong et al (2018) during this peer review. We appreciate the scientific contribution of Zong et al.

For the comment regarding the statement “This approximation correctly captures qualitative features away from the IC-CDW transition” we have missed the word “away from”. Also, by “random” we meant “fluctuating”. Our apologies for this confusion. However, the problem with diffuse IC-CDW peaks still remains.

For example, the authors write “The IC peaks we observe in 1T-TaS2 match that of all previous reports” but we cannot agree on this statement. The IC peaks obtained by the authors are indeed diffuse. Diffuse peaks are really extended, which is typically due to structural imperfections. But if the satellite peak size is comparable to the primitive peaks from TaS2 lattice, then the peaks are spot-like and not diffuse. Spot-like peaks mean long-range order of more than 100 layers or so.

Robert Hovden, PhD
Assistant Professor
Materials Science & Eng.
University of Michigan
hovden@umich.edu
hovdenlab.com
c. 770-265-4042

We respectfully disagree with the Reviewer's interpretation of our diffraction patterns in Fig 2a. Superlattice peaks of IC phase (Fig. 2a, right) are clearly diffuse. For comparison, the superlattice peaks at lower temperatures (Fig. 2a left and bottom) are sharp which provides a baseline for what a 'spot-like' diffraction peak is. Here, one should not directly compare the width of superlattice peaks to Bragg (primitive) peaks, as the Bragg peaks are orders of magnitude larger in intensity and saturate the image.

We can find spot-like IC-CDW peaks in 1T-TaS₂, such as Figure 2 in [G. Storeck et al., Structural Dynamics 7, 034304 (2020)] and Fig. 2b) and Fig. 3 c) in [Vogelgesang_2017] which was cited by the authors. Therefore, we find that the authors' discussion on the IC phase is inconsistent with previous reports.

For these reasons, it is difficult to believe that the authors' free energy theory qualitatively reproduces diffraction patterns.

We revise the quoted response in our previous letter to read, "The IC peaks we observe in 1T-TaS₂ match those previous reports". The word "all" has been deleted and "that" is changed to "those". We acknowledge the discrepancy in diffraction patterns the author has referenced but note the references [G. Storeck et al., Struct. Dyn. 7, 034304 (2020)] and [S. Vogelsang et al., Nature Physics 14, 184 (2018)] describe non-equilibrium states. Our results are expected for IC diffraction measurements and were reproducible over long periods of time (multiple days). We do not consider transient behavior nor optical excitations. Our Monte-Carlo simulations qualitatively recreate the IC and twinned CDW diffraction peaks we measure—we now make this more explicit with an inset to Figure 3g.

Major Comment 5:

The authors do not understand the NC phase in 1T-TaS₂. They argue that the NC phase is a disordered phase. However, this argument is wrong for the system considered in this manuscript (namely TaS₂ at equilibrium). The authors give references to refute this comment, but these references are not appropriate: They consider different materials (square lattice) or optically-excited/transient systems. In fact, this problem of citing inappropriate references appears repeatedly in the manuscript. We commented on this issue but it was not addressed. (This problem appears again in the new reply. The authors clearly write "we do not consider transient behavior nor optical excitations" but they try to support their argument by citing work on optically-excited/transient systems).

Whether the NC phase is described as disordered now has little relevance as we removed any description of NC-CDW as disordered (in response to Reviewer #3's previous comments). The NC phase in 1T-TaS₂ remains contentious and readers are best served by investigating these details throughout the literature.

Robert Hovden, PhD
Assistant Professor
Materials Science & Eng.
University of Michigan
hovden@umich.edu
hovdenlab.com
c. 770-265-4042

We are aware of the work by Gerasimenko et al. about the so-called Hidden (H) CDW state which appear at highly non-equilibrium condition. They report the appearance of chiral order in a seemingly disordered H phase. In this reference, Fig. 2c (NC phase) is cited as “At elevated temperatures, $T > T_{\{C-NC\}} \sim 220$ K, the uniform C order changes to a modulated nearly commensurate (NC) CDW with a highly regular array of domains”. Here, “highly regular array of domains” is used in contrast to the disordered domain-walls of the H-CDW phase. Please understand the references before citing them: It is not the work of the referees to do so. Therefore, the authors write “At room temperature, the conductive NC-CDW is generally accepted as a C-CDW with short range order [Gerasimenko et al., ...] ...” but this is incorrect. The NC-CDW phase and the H-CDW phase are different CDW phases.

We agree with the Reviewer that the main point of [Gerasimenko et al] primarily pertains to the H-CDW state. However, we have a disagreement as to whether the NC phase measurements reported by Gerasimenko (their Fig. S3b and 2c) should be described as ordered or disordered. As stated above, this is avoided as we no longer refer to the NC phase as disordered. We do write, “At room temperature, the conductive NC-CDW is generally accepted as a C-CDW with short range order”. We think this statement is justified but have added support by a recent STM/DFT work [J. W. Park, *Nat. Commun.* 10, 4038 (2019)].

We have reviewed all of our references and made improvements as follows:

Pg. 1 para 1 now reads:

“Charge density waves (CDW) are an emergent periodic modulation of the electron density that permeates a crystal with strong electron-lattice coupling [Wilson_1975, Chan_1973, Hellmann_2012, Pillo_2020]. TaS_2 and $\text{TaS}_x\text{Se}_{2-x}$ host several CDWs that spontaneously break crystal symmetries, mediate metal-insulator transitions [Wilson_1975, Hellmann_2012], and compete with superconductivity [Wilson_1975, Navarro_2016, Ang_2015, Li_2017].

[Wilson_1975]: Wilson et al., *Adv. Phys.* **24**, 117–201 (1975)

[Chan_1973]: Chan et al., *J. Phys. F: Met. Phys.* **3**, 795 (1973)

[Hellmann_2012]: Hellmann et al., *Nat. Commun.* **3**, 1069 (2012)

[Pillo_2000]: Pillo et al., *Phys. Rev. B* **62**, 4277 (2000)

[Navarro_2016]: Navarro-Moratalla et al., *Nat. Commun.* **7**, 11043 (2016)

[Ang_2015]: Ang et al., *Nat. Commun.* **6**, 6091 (2015)

[Li_2017]: Li et al., *npj Quantum Mater.* **2**, 11 (2017)

Pg. 1 para 3 now reads:

An intermediate triclinic phase has also been reported [Tanda_1985, Nakatsugawa_2020, Wang_2019, Gerasimenko_2019, Coleman_1988]. At room temperature, the conductive NC-CDW is generally accepted as a C-CDW with short range order [Gerasimenko_2019, Coleman_1988,

Robert Hovden, PhD
Assistant Professor
Materials Science & Eng.
University of Michigan
hovden@umich.edu
hovdenlab.com
c. 770-265-4042

Wu_1989, Vogelgesang_2017, Hovden_2016] that permits electron transport along regions of discommensuration [Cho_2016, Sipos_2008, **Park_2019**].

[Tanda_1985]: Tanda et al., *Synth. Met.* **11**, 85–100 (1985)

[Nakatsugawa_2020]: Nakatsugawa et al., *Sci. Rep.* **10**, 1239 (2020)

[Wang_2019]: Wang et al., *Sci. Rep.* **9**, 7066 (2019)

[Gerasimenko_2019]: Gerasimenko et al., *npj Quantum Mater.* **4**, 32 (2019)

[Coleman_1988]: Coleman et al., *Adv. Phys.* **37**, 559-644 (1988)

[Wu_1989]: Wu et al., *Science* **243**, 1703–1705 (1989)

[Vogelgesang_2017]: Vogelgesang et al., *Nat. Phys.* **14**, 184–189 (2017)

[Hovden_2016]: Hovden et al., *Proc. Natl. Acad. Sci.* **113**, 11420–11424 (2016)

[Cho_2016]: Cho et al., *Nat. Commun.* **6**, 10453 (2016)

[Sipos_2008]: Sipos et al., *Nat. Mater.* **7**, 960–965 (2007)

[Park_2019]: Park et al., *Nat. Commun.* **10, 4038 (2019)**

Conclusion: We feel that there is a serious problem of scientific consistency in this manuscript. Also, many comments were not addressed properly. Therefore, we must reject this manuscript.

We believe this experimental work is accurately reported and impactful. A majority of our manuscript is now in agreement with the Reviewer although respectful disagreement remains on minor aspects. We want to sincerely thank the Reviewer for many comments and inquiries that have strengthened the quality of our work and how it is presented.

We have addressed each of the comments from the Reviewer. The changes have improved the manuscript. Please do not hesitate to contact us with any further comments or requests.

Sincerely,

Robert Hovden, Ph.D.